# *Azolla filiculoides* L. as a source of metal-tolerant microorganisms

**Artur M. Banach**[1]*, **Agnieszka Kuźniar**[1], **Jarosław Grządziel**[2], **Agnieszka Wolińska**[1]

**1** Department of Biology and Biotechnology of Microorganisms, The John Paul II Catholic University of Lublin, Lublin, Poland, **2** Department of Agricultural Microbiology, Institute of Soil Science and Plant Cultivation, State Research Institute, INCBR Centre, Puławy, Poland

* abanach@kul.pl

**Data Availability Statement:** The identified sequences were deposited in the GenBank (NCBI) under the following accession number: PRJNA589741 available at https://www.ncbi.nlm. nih.gov/bioproject/589741.

## Abstract

The metal hyperaccumulator *Azolla filiculoides* is accompanied by a microbiome potentially supporting plant during exposition to heavy metals. We hypothesized that the microbiome exposition to selected heavy metals will reveal metal tolerant strains. We used Next Generation Sequencing technique to identify possible metal tolerant strains isolated from the metal-treated plant (Pb, Cd, Cr(VI), Ni, Au, Ag). The main dominants were Cyanobacteria and Proteobacteria constituting together more than 97% of all reads. Metal treatment led to changes in the composition of the microbiome and showed significantly higher richness in the Pb-, Cd- and Cr-treated plant in comparison with other (95–105 versus 36–44). In these treatments the share of subdominant Actinobacteria (0.4–0.8%), Firmicutes (0.5–0.9%) and Bacteroidetes (0.2–0.9%) were higher than in non-treated plant (respectively: 0.02, 0.2 and 0.001%) and Ni-, Au- and Ag-treatments (respectively: <0.4%, <0.2% and up to 0.2%). The exception was Au-treatment displaying the abundance 1.86% of Bacteroidetes. In addition, possible metal tolerant genera, namely: *Acinetobacter*, *Asticcacaulis*, *Anabaena*, *Bacillus*, *Brevundimonas*, *Burkholderia*, *Dyella*, *Methyloversatilis*, *Rhizobium* and *Staphylococcus*, which form the core microbiome, were recognized by combining their abundance in all samples with literature data. Additionally, the presence of known metal tolerant genera was confirmed: *Mucilaginibacter*, *Pseudomonas*, *Mycobacterium*, *Corynebacterium*, *Stenotrophomonas*, *Clostridium*, *Micrococcus*, *Achromobacter*, *Geobacter*, *Flavobacterium*, *Arthrobacter* and *Delftia*. We have evidenced that *A. filiculoides* possess a microbiome whose representatives belong to metal-resistant species which makes the fern the source of biotechnologically useful microorganisms for remediation processes.

## Introduction

Plants are one of the most important hosts for complex communities of microorganisms colonizing plant tissues (endosphere–endophytes), outer plant surfaces (phyllosphere–epiphytes) and root surfaces (rhizosphere) forming the plant microbiota [1]. Microbes in these niches can establish beneficial, neutral (commensalism) or detrimental (parasitism or pathogenicity) associations of varying intimacy with their host plants. Specific interactions between microbes

**Funding:** The authors received no specific funding for this work.

**Competing interests:** The authors have declared that no competing interests exist.

and model plants, such as in *Rhizobium*-legume symbioses [2], are well understood but the majority of the plant microbiome and its contribution to the extended phenotype of the host is not yet well defined. Importantly, the microbiome is strongly influenced by the plant genome and may be considered as an extension forming a second genome or collectively forming a pan-genome [3,4]. As the plant microbiome is a key determinant of plant health and productivity [5], it has received substantial attention in recent years [6,7].

Plant-associated microorganisms produce substances such as phytohormones and antibiotics which support plant growth and provide protection against pathogens (Plant Growth Promoting Bacteria, PGPB) [4,8,9]. This property can help improve crop production–one of today's most important issues due to rising human population and the shortage in resources [10]. In addition, these microorganisms can help plants to survive stress conditions, e.g. drought, salinity [11] or the presence of pollutants [12].

Heavy metals are one of the most important environmental pollutants. Unlike organic substances, metals cannot be degraded by biological or chemical means and this makes them very hazardous substances that can have harmful effects on ecosystems [13,14]. Lead, cadmium, chromium and nickel belong to the most widespread heavy metals in environment and are considered as non-essential (highly toxic) trace elements. Because of their long persistence in environment they tend to accumulate in soils, migrate to waters both entering into food chain resulting in concentrations exceeding safety limits which eventually leads to serious health consequences and deterioration soil productivity [13,15]. Noble metals such as gold and silver are also common in environment being important part of jewelry and electronics industry [16] and also belong non-essential trace elements having negative effects on living organisms [13,15]. The structure of soil microbial populations is also related to the levels of heavy metals. As a result of metal presence microbial metabolism is affected resulting in lower soil health and fertility [14]. The response of microorganisms depends not only on concentration and availability of metals but also on type of metal, medium and the species present [15].

Due to above mentioned problems there is a need for metal removal from contaminated environments. There are several techniques for coping with heavy metal pollution including chemical precipitation, oxidation or reduction, filtration, ion-exchange, reverse osmosis, membrane technology, evaporation and electrochemical treatment. But most of these techniques become ineffective or expensive when metal concentrations are below 100 mg L$^{-1}$. In addition, good solubility of metals salts in water make them impossible to be separated with physical methods [15]. Biological methods allowing the use of plant and microorganisms for metal biosorption and/or accumulation are an attractive, efficient and environmentally friendly alternative to physical/chemical methods [13,15]. Microorganisms possess many adaptation allowing them to detoxify metals via biosorption, bioaccumulation, biotransformation and biomineralization what is used for different bioremediation methods. Similarly to it plants can be also used for metal remediation which is also very effective solar energy-driven method for metal removal (phytoremediation). The plants having ability to accumulate high levels of metals and growing very fast are called hyperaccumulators and are the best for this purpose. However, there is limited number of such plants and the solution for this may be an aid of microorganisms. As they possess many mechanisms to cope with heavy metals and, as mentioned above, are able to support plants growth (PGPB) their application may be beneficial for successful phytoremediation of metals [15]. For that reason, studying microbiomes with regard to metal tolerance is of interest for better designing metal-treatment processes. The process of supporting plants in phytoremediation is called assisted phytoremediation [12,17]. Microorganisms involved in this process are well-adapted to metal-polluted environments, where higher organisms are unable to occur. They have developed capabilities to protect

themselves from heavy metal toxicity by various mechanisms such as adsorption, uptake, methylation, oxidation, and reduction [18].

Among hyperaccumulators effective in heavy metal removal by phytoremediation [19] is the aquatic fern *Azolla filiculoides* L. (Salviniaceae), which possess a recently recognized endophytic microbiome [20,21] composed of interesting microorganisms with PGPB potential [21]. The plant was tested in numerous studies for the potential of the removal many metals among which Pb, Cd, Cr, Ni, Ag and Au are often studied [19,22–25]. However, the tolerance of *A. filiculoides* to heavy metals such as Pb, Cd, Cr, Ni, Ag and Au and possible role in assisted phytoremediation remains unknown. To answer the first question, we exposed *A. filiculoides* to specific metals (Pb, Cd, Cr(VI), Ni, Ag and Au); the removal efficiency of which by the fern had been tested in the past [22] and their dosage were based on literature information about *Azolla* sp. tolerance to these metals. We used the non-treated plant (NTP) as a reference. The Next Generation Sequencing (NGS) technique was applied to recognize changes in the *A. filiculoides* microbiome composition under heavy metal stress. It was hypothesized that the exposition of *A. filiculoides* to Pb, Cd, Cr(VI), Ni, Au(III) and Ag will affect the microbiome structure depending on metal. The resulting microbiome composition will display various microbial groups of different tolerance to selected metals and the most abundant groups would be identified as metal tolerant species.

## Materials and methods

### Plant material

*Azolla filiculoides* L., serving as a source of microorganisms, originated from our laboratory culture established in 2010 using material obtained from Warsaw Botanical Garden (Poland). The plant was cultivated according to the International Rice Research Institute recommendation [26]. The same medium and conditions were used for the heavy metal stress experiment.

### Exposition to heavy metals

Plants were grown in 1 L closed glass containers on IRRI medium. The following metals and dosages were applied to the IRRI medium: $Pb(NO_3)_2$ 500 mg $L^{-1}$, $Cd(NO_3)_2$ 5 mg $L^{-1}$, $K_2Cr_2O_7$ 50 mg $L^{-1}$, $NiCl_2$ 50 mg $L^{-1}$, $H[AuCl_4]$ 5 mg $L^{-1}$, $AgNO_3$ 5 mg $L^{-1}$ making 6 treatments. A non-metal threated *A. filiculoides* culture served as control. Metal doses were selected as follows: Pb 500 mg $L^{-1}$, Cd 5 mg $L^{-1}$, Cr(VI) 100 mg $L^{-1}$, Ni 100 mg $L^{-1}$, Au(III) 5 mg $L^{-1}$ and Ag 5 mg $L^{-1}$ [27–30]. The selection of metals was based on our previous studies with *Azolla* sp. [22]. These metals constitute environmental risk and their mitigation is often studied. Plants were incubated in 16/8 h photoperiod (3500 lux light energy, fluorescent lamps Philips Master TL-D 36W/830) at 20.69 ± 1.55˚C, and relative humidity of 84.5 ± 5.16% (H-881t hygrometer, Zootechnika, Poland). All samples were exposed to metals for one week which we thought to be sufficient to affect the microbiome of *A. filiculoides*. After this time plant material was collected and used for isolation of endophytic microorganisms.

All reagents were dedicated for microbiological analyses and purchased from Sigma-Aldrich; water was deionized and sterilized before use (sdH$_2$O).

### Isolation of microbial DNA

Three portions of each sample (2 g) were randomly selected in order to provide repeatability. Next, the samples were washed several times in distilled water to remove the IRRI medium. The main procedure was proceeded by sterilization of all materials in a laminar chamber. After that, the plants were immersed for a given time in subsequent reagents: (1) 96% ethanol

for 60 s, (2) 3% NaClO for 6 min, (3) 75% ethanol– 60 s, and (4) 3 times 1 min in fresh sdH$_2$O. The efficiency of sterilization was assessed by inoculating Petri dishes with the water from the last washing. Each sterilized sample was ground in an ice-cooled mortar using 1 mL 12.5 μM phosphate buffer. The macerate was centrifuged at 3000 × g (5 min, 4˚C) and three 300-μL samples of each supernatant were transferred into Eppendorf tubes for further DNA isolation.

Microbial DNA from sterilized plants was isolated using the PowerLyzer® PowerSoil® DNA Isolation Kit (MO BIO Laboratories Inc., QIAGEN, USA) according to the manufacturer's instructions. Next, the obtained DNA samples were pooled [31,32] and used for PCR reaction. The PCR mixture contained 5x FIREPol® Master Mix (Solis BioDyne, Estonia), 1 μL of template DNA, and sterile double-distilled water (free DNase) in a total volume of 20 μL. Universal eubacterial primers, targeting the hypervariable V3-V4 region of the 16S rRNA gene (each 1.0 μM): 27F (5′-AGAGTTTGATCATGGCTCAG-3′) and 518R (5′-GTATTACCGCG GCTGCTGG-3′) were used [33]. The reaction was carried out under the following conditions: 98˚C for 10 s; 30 cycles of 95˚C for 5 s, 56˚C for 5 s, and 72˚C for 40 s (LABCYCLER, Senso-Quest GmbH, Germany). The PCR products were run on agarose gel (1%) and visualized with the use of SimplySafe® (EURx, Poland). Additionally, control reactions were performed: negative–containing only sterile double-distilled water (free DNase) without a DNA template and positive, in which DNA isolated from E. coli DH5α™ was a template. Then, all PCR products were purified and sent for sequencing using Illumina MiSeq 2000 NGS technology (Genomed S.A., Poland).

## Data processing

The raw reads from Illumina sequencing were further processed using R language environment [34,35] and DADA2 package [36]. Sequences were trimmed to 250 bp, and the first 20 bp were removed (comprising primers and low-quality bases) from both read directions. The maximum number of "N" bases was set to 0; the maximum expected error was set to 3; and the sequences were truncated at the first instance of a quality score lower than or equal to 2 (which corresponds to 0.63% probability that the base is incorrectly assigned). Other parameters were set to default. The sequences were dereplicated with default parameters and exact sequence variants (ASVs) were resolved. Next, chimeric sequences were removed using the consensus method. Taxonomy was assigned using the latest version of the RDP database (release 11.5) using a naïve Bayesian classifier [37] and the minBoot parameter set to 80. The resulting taxonomy and read-count tables constructed in DADA2 were appropriately converted and imported into the phyloseq package [38]. Alpha diversity measures were calculated with the use of the phyloseq package.

A Principal Component Analysis (PCA) plot was generated in STAMP using Welsch's two-sided t-test and the significant differences between the A and B groups were tested with a t-test [39]. Significance was assumed at p-value below 0.05.

The identified sequences are available under accession number PRJNA589741 in the GenBank database (NCBI, https://www.ncbi.nlm.nih.gov/bioproject/589741).

## Results

The performed NGS resulted in various numbers of reads depending on the sample studied: the highest numbers of reads were obtained for the non-treated plant (196,483) followed by 160,543 (+Cr) and 146,217 (+Pb). Exposition of *A .filiculoides* to Ag(I) and Ni(II) resulted in 122,670 and 121,541 reads, respectively, and the lowest numbers of reads were noted for Au (III)– 113,063 and Cd(II)– 110,190.

## Archaea and bacteria

Cyanobacteria and Proteobacteria seemed to be the two main dominants in all samples constituting together more than 97% of all reads. Next, as subdominants: Bacteroidetes (up to 1.8% in +Au), Firmicutes (up to 0.87% in +Cr) and Actinobacteria (0.76%, +Cr) were identified. The remaining of the obtained reads, constituting <0.1% were grouped into the "Other" cluster constituting up to 0.12%. After NGS analysis, we established that 0–0.16% of reads belonged to Archaea (Fig 1). Euryarchaeota were only represented by the two following genera: *Methanobacterium* and *Methanoregula*. More precisely, *Methanobacterium* sp. was detected in the non-treated plant (NTP) (0.003%), +Pb (0.027%), +Cr (0.15%) and +Ag (0.005%) treatments, while *Methanoregula* sp. only in the Cr-treated *A. filiculoides* (0.013%).

## The cyanobiont of *A. filiculoides*

Cyanobacteria were represented only by *Anabaena azollae* 0708, the main known symbiont of *A. filiculoides*. Its share depended on the metal used: a lower abundance of *A. azolla*e by 6.24, 12.55 and 12.75% was observed for nickel, chromium and gold, respectively, in comparison to NTP (48.77%). Exposition to other metals led to slightly higher *A. azollae* percentages of 50.99 (lead), 52.24 (silver) and 53.55% (cadmium) (Fig 1).

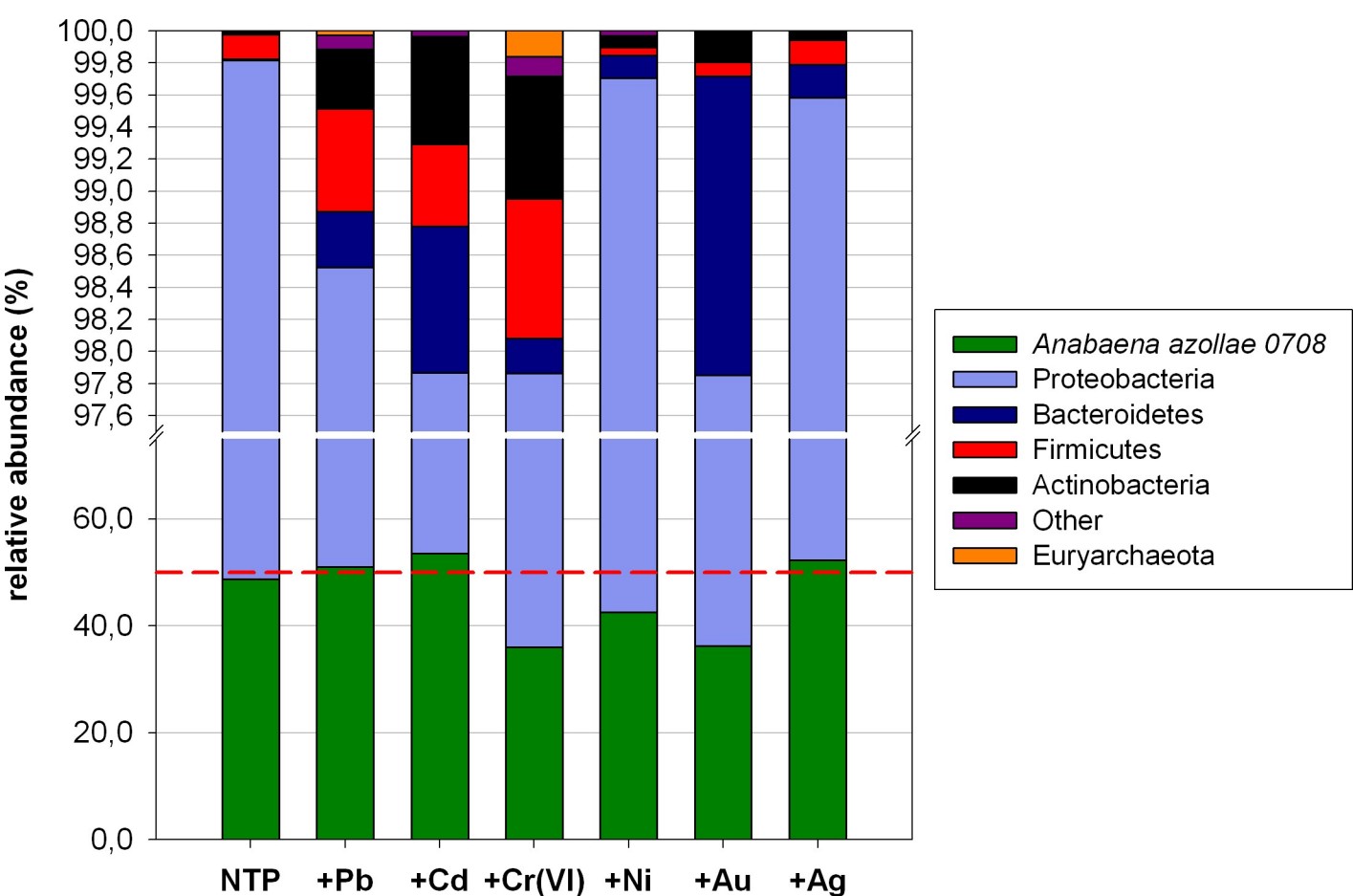

**Fig 1. The relative abundance of bacteria at the phylum level across all samples.**

## Dominant bacterial phyla

Proteobacteria constituted 51.05% of the microbiome of the non-treated *A. filiculoides*. A lower percentage of these bacteria was noted for Cd (44.31%), Ag (47.34%) and Pb (47.54%), whilst a higher abundance was recorded for Ni- (57.18%), Au- (61.64%) and Cr-treated (61.85%) microbiomes (Fig 1). When the share of Cyanobacteria to Proteobacteria (Cy:Pr ratio) were compared, it was stated that Cr, Ni and Au have the strongest effect by hampering *A. azollae* and promoting Proteobacteria (Cy:Pr 0.58, 0.74 and 0.59, respectively). In the presence of Cd, Ag and Pb, the ratios of 1.21, 1.10 and 1.07, respectively, suggested dominance of Cyanobacteria over Proteobacteria. NTP were characterized by a ratio of 0.96 suggesting an almost equal share of both phyla (Fig 1).

In the case of Cyanobacteria the species name is given. The phyla termed 'Other' group bacteria constituting <0.1% of the total reads. The dotted red line presents the Cy:Pr ratio of 1.0.

Subdominated Bacteroidetes reached the lowest abundance in NTP (0.0088%). Exposition of microbiome to heavy metals resulted in a noticeable percentage of Bacteroidetes in the total microbial pool. Their abundance was as follows: +Pb: 0.347%, +Cd: 0.91%, +Cr: 0.225%, +Ni: 0.14%, +Au: 1.86% and +Ag: 0.21%.

Firmicutes share was the lowest in +Ni (0.05%) and Au (0.09%) followed by NTP and +Ag (both 0.15%). In contrary, the highest percentage was noted in +Cd (0.51%), +Pb (0.64%) and +Cr (0.87%).

Actinobacteria constituted 0.018% of the total microbiome in NTP, 0.048% in Ag-, 0.072% in Ni-, 0.19% in Au-, 0.37% in Pb-, 0.68% in Cd- and 0.76% in Cr-treatment.

Other phyla, referred to as 'rare', constituting <0.1% of the total microbiome were represented by: Armatimonadetes, Caldiserica, Chlamydiae, Chloroflexi, Deinococcus-Thermus, Fusobacteria, Gemmatimonadetes, Nitrospirae, Planctomycetes, Spirochaetes and Verrucomicrobia. We did not detect any of them in Au-treatment and a very low share of 0.005 and 0.003% in +Ag and NTP, respectively. Ten times higher abundance was noted in +Ni (0.031%) and +Cd (0.034%) and highest in +Pb (0.089%) and +Cr– 0.12%.

## Proteobacteria structure – classes and genera level

Alphaproteobacteria were the most abundant class of Proteobacteria constituting 63–99% (Fig 2). The highest abundance was noted in NTP (98.56%), followed by +Cd (94.97%), +Cr (93.45%) and +Ag (90.5%). Other metals strongly affected the percentages of this class– 74.53% in Pb-, 66.35% in Ni- and 62.65% in the Au-treated microbiome. Simultaneously, the share of Beta- and Gammaproteobacteria increased in these three samples. In non-treated plant Beta- and Gammaproteobacteria constituted 0.3996% and 1.03%, respectively, +Cd, +Cr and +Ag treatments showed 3.77, 5.54, 2.65% (β-Proteobacteria), 1.197%, 0.84, 6.83% (γ-Proteobacteria), respectively. Pb, Ni and Au treatments had β-Proteobacteria at the level of 12.29, 15.57 and 5.34%, respectively. The abundance of γ-Proteobacteria in these treatments was as follows: 13.14, 18.07 and 32%, respectively. Deltaproteobacteria were hardly detected in the samples studied, their abundance ranged between 0.0098% (+Ag) through 0.0108% (NTP), 0.0115% (+Ni) to 0.053% (+Pb), 0.065% (+Cr) and 0.168% (+Cr). There were no Deltaproteobacteria in gold-treated *A. filiculoides* (Fig 2).

Three dominant genera were recorded among Alphaproteobacteria–*Brevundimonas*, *Rhizobium* and *Asticcacaulis* (Fig 3). In NTP, they constituted 63.46, 24.49 and 8.39%, respectively. After microbiome exposition to metal contamination, the abundance of *Brevundimonas* sp. declined in all treatments in the following magnitudes: Cr– 47.55%, Ag– 41.56%, Ni– 32.5%, Pb– 24.37%, Au– 16.59% and Cd– 8.7%. The abundance of the two remaining dominants depended on the metal; for *Rhizobium* sp. it was higher for Cd (77.8%), Pb (34.4%) and Ag

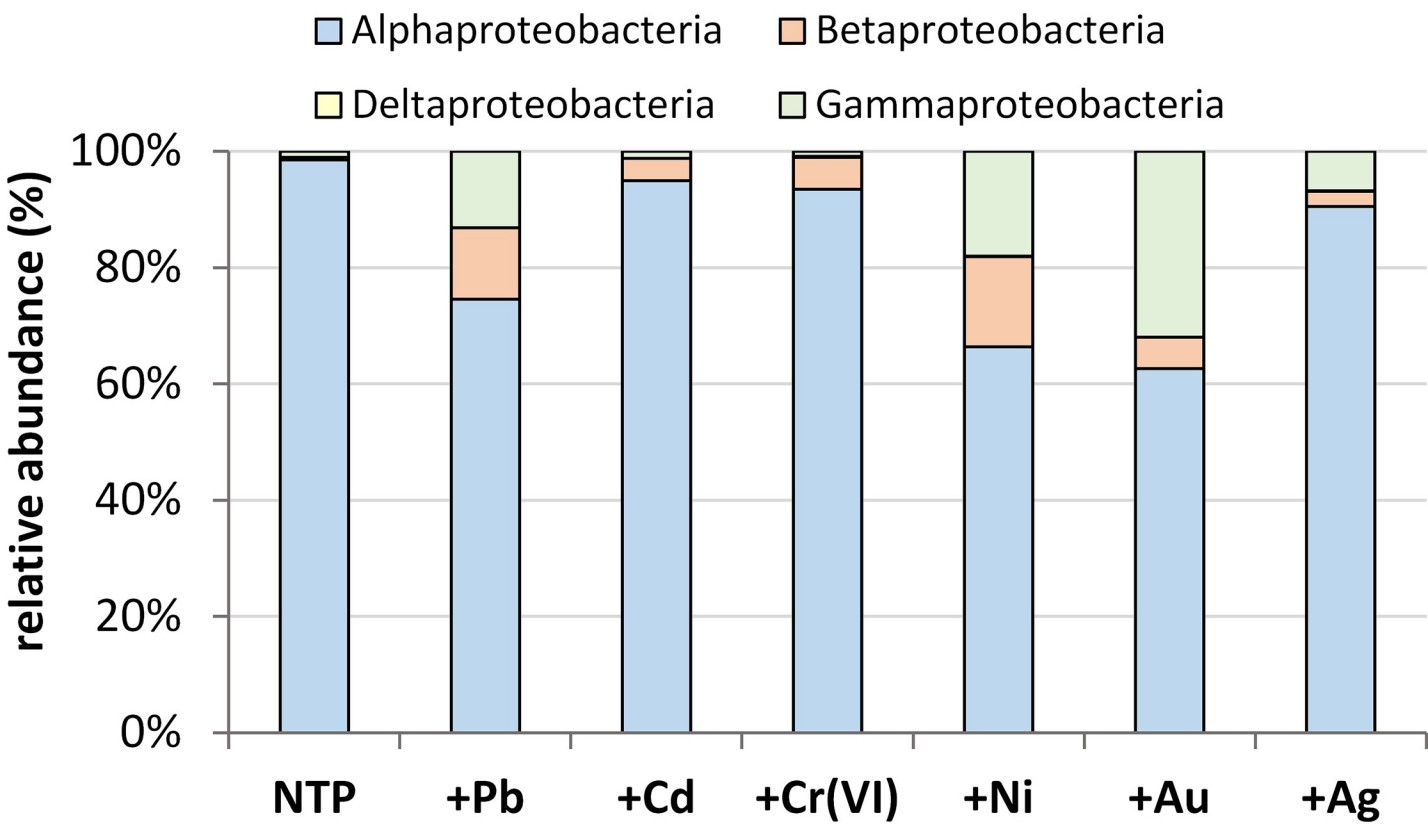

**Fig 2. Decomposition of proteobacteria on 4 classes in analysed samples.**

(33.66%), while exposition to Ni, Au and Cr resulted in a drop in *Rhizobium* sp. percentages, respectively: 21%, 20.4% and 14.2%. For *Asticcacaulis* sp. we recorded a 12–25% increase in almost all treatments but a much lower increase for cadmium (7%). The remaining 23 detected genera, constituting <1% of Proteobacteria, were grouped together in one (for all Proteobacteria classes) 'Other' cluster (S1 Table). The 'Unclassified' cluster was composed of representatives of unidentified microorganisms belonging to all Proteobacteria classes. In the case of α-Proteobacteria, 10 families with non-recognized microorganisms were recorded, constituting up to 1.79% (S5 Table).

In β-Proteobacteria, two dominants, *Burkholderia* sp. and *Methyloversatilis* sp., were distinguished, both showing very low abundances in NTP of 0.054 and 0.022% respectively. Exposition of the microbiome to the metals studied led to much higher percentages of both dominants. The abundances of the genus *Burkholderia* were 11.7% (+Pb), 0.9% (+Cd), 5.1% (+Cr), 15.3% (+Ni), 5% (+Au) and 2% in +Ag (Fig 3). *Methyloversatilis* constituted 0.34% (+Pb), 1.87% (+Cd), 0.32% (+Cr), 0.08% (+Ni), 0.2% (+Au) and 0.5% (+Ag) of β-Proteobacteria. Twelve rare genera were clustered into the 'Other' group constituting up to 0.6% of β-Proteobacteria (S2 Table). Unidentified microorganisms, belonging to 4 families, constituted 0.3% of this class (S5 Table).

Within Deltaproteobacteria, only *Geobacter* sp. was present in more than one sample including NTP (0.011%). Exposition to lead, chromium and gold lead to an increase of its abundance of 0.05, 0.089, and 0.012%, respectively (Fig 3). We also recorded the presence of 5 other genera with a percentage below 0.01% (S3 Table) and unrecognized microorganisms belonging to 4 families (S5 Table).

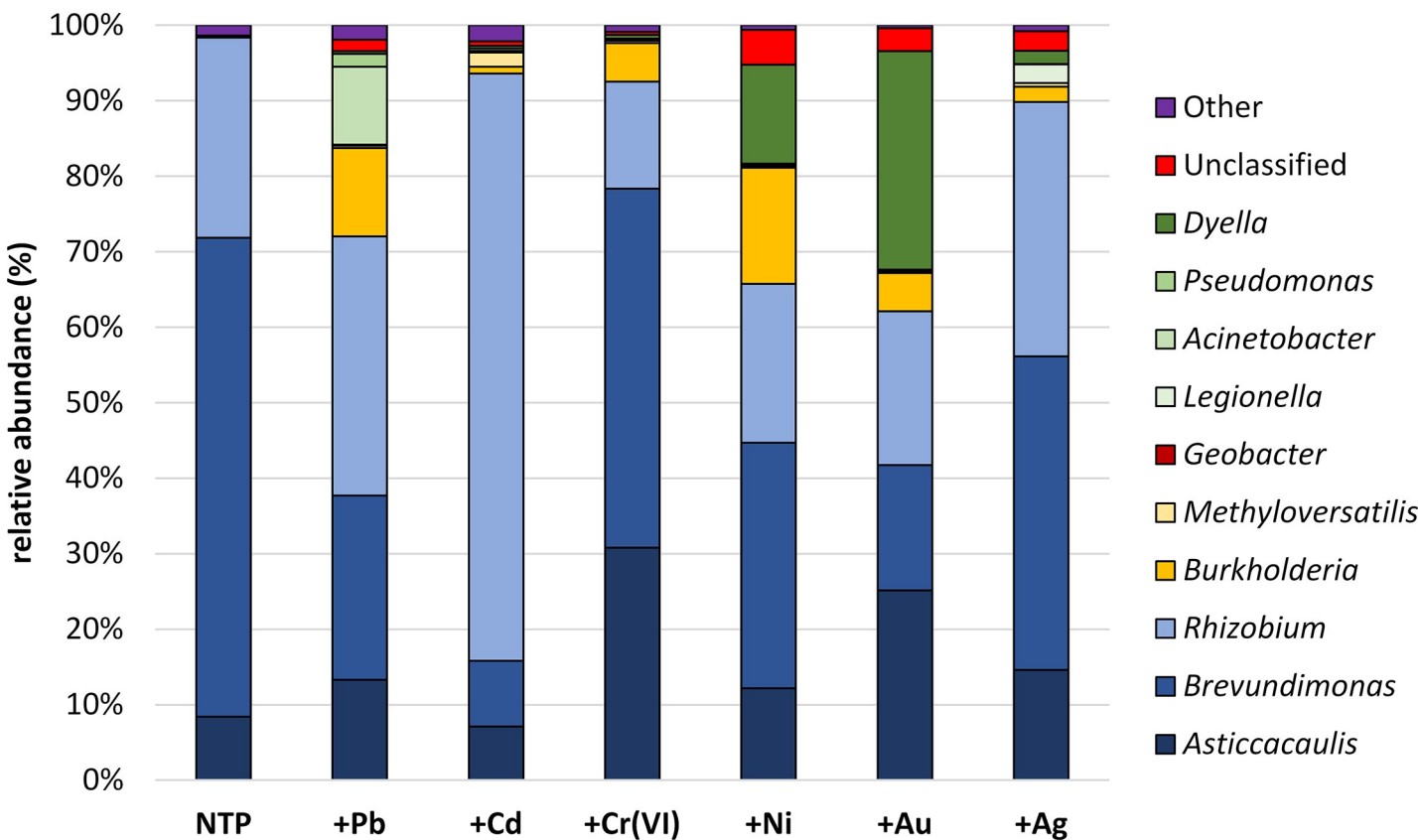

**Fig 3. The main dominant genera of proteobacteria separated into α- (blue), β- (yellow), δ- (brown) and γ-classes (green).** Violet color represents rare (<1%) 58 genera from all classes grouped together (S1–S4 Tables) and in red all unclassified microorganisms (22 families)–S5 Table.

Gammaproteobacteria were notably abundant in Ni, Au and Ag treated *A. filiculoides* (Fig 3). We distinguished 4 dominant genera: *Legionella*, *Acinetobacter*, *Pseudomonas* and *Dyella*. Their abundances were below 0.1% and *Legionella* sp. was absent in the non-treated material. The highest percentages were noted for *Dyella* sp. which amounted to 28.9% in +Au, 13% +Ni and 1.785% in +Ag, while ranging between 0.33–0.46% in other metal treatments. *Acinetobacter* sp. was the most abundant in Ni-treatment (10.35%) where it constituted 0.1–0.3% and in Cr-Ni-Cd treatments, followed by <0.1% abundances in the noble metal treatments. *Legionella* sp. was most common in +Ag (2.49%), followed by +Cd (0.2%), +Au (0.18%) and <0.1% for other metal treatments. The 15 genera with the lowest abundance (<0.1%) were grouped into one cluster (S1D Table). Their share was highest in NTP (0.89%). The proportion of unidentified microorganisms (4 families, S5 Table) was noticeable in +Ni (4%), +Au (+2.8%) and +Ag (2.48%).

## Dominant genera among the less abundant bacterial phyla

Bacteroidetes evidenced a huge difference in bacterial composition between treated and non-treated samples (Fig 4). The non-metal treated plant microbiomes were composed only of *Chryseobacterium* sp. (75%) and *Flavobacterium* sp. (25%) (Flavobacteriaceae). Other genera become detectable when *A. filiculoides* was exposed to the studied metals. The presence of *Flavobacterium* sp. was also confirmed in Pb- and Cr-treated microbiomes reaching a share of 12.55 and 2%, respectively. *Sediminibacterium* sp. was present in all metal-treated

# Bacteroidetes

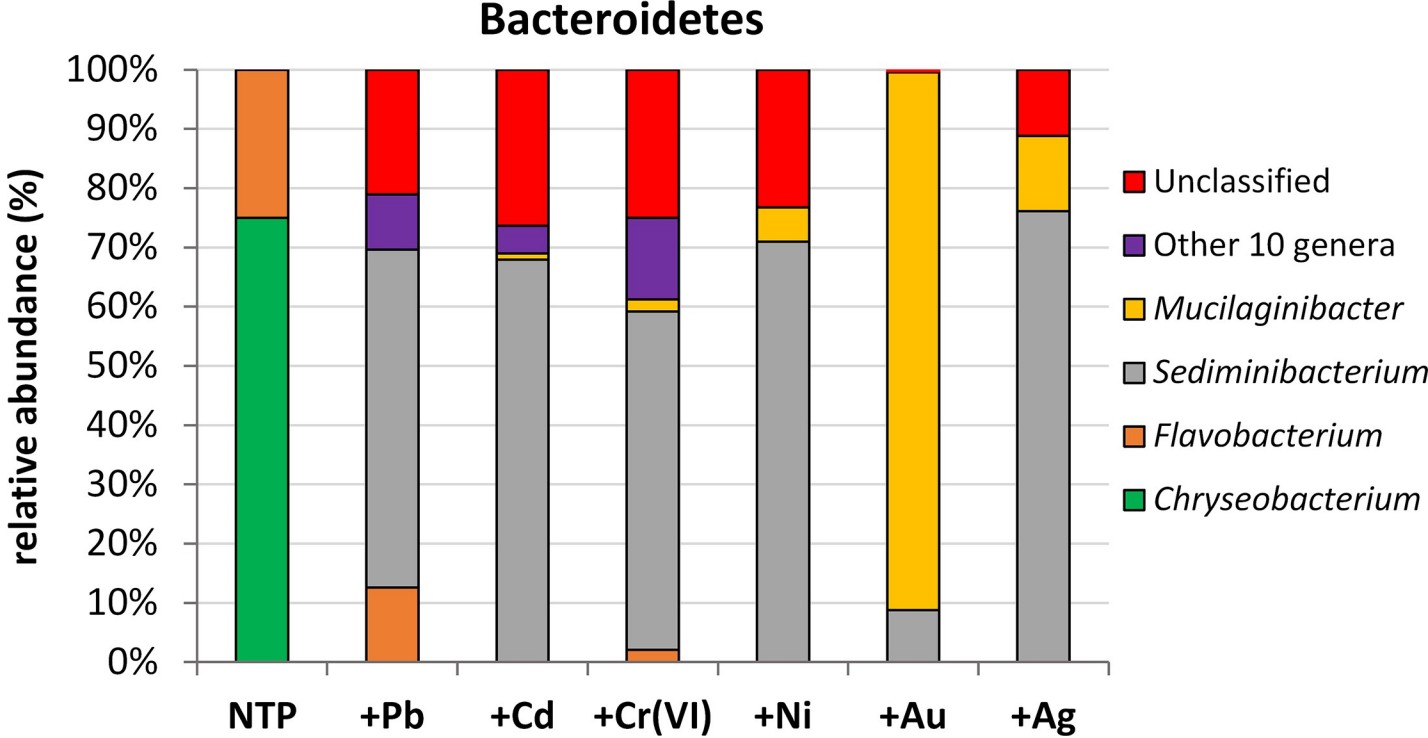

**Fig 4. The main genera dominants within bacteroidetes.** The 'Other' group represents those with abundance <10%.

microbiomes constituting: 57% in +Pb and +Cr, 67.9% in +Cd, 70.9% in +Ni and 76% in +Ag treatment. Only exposition to Au led to much a lower percentage of this genus (8.75%). The fourth dominant genus was *Mucilaginibacter* showing this highest abundance of 90.79% in +Au treatment. Ag treatment also revealed the presence of this genus at a level of 12.69%. The presence of *Mucilaginibacter* sp. was also detected in +Cd, +Cr and +Ni but at a much lower level– 1.1, 2 and 5.8%, respectively. There was no *Mucilaginibacter* sp. in Pb-treated material. Due to the proportion of Bacteroidetes of up to 2% in the whole microbiome we grouped other genera with abundance of less than 10% into one cluster. These genera constituted 0.09% of the total microbiome and was composed of *Ferruginibacter*, *Vitellibacter*, *Prevotella*, *Pedobacter*, *Rhodocytophaga*, *Bacteroides*, *Hymenobacter*, *Spirosoma*, *Flavisolibacter* and *Alistipes* genera (S6 Table). The highest abundance of this group of 13.76% was recorded in Cr treatment. Pb-, Cd- and Au-treatment contained 9.3, 4.6 and 0.18% of these genera. There were none of them in the +Ni and +Ag samples (Fig 4). Unclassified bacteria constituted an important fraction of Bacteroidetes ranging between 11.19 (+Ag) and 21–26.38% (heavy metals). Au treatment had only 0.273% of unclassified reads. The highest abundance, constituting 61–100% of all unidentified microorganisms from Bacteroidetes, showed unclassified representatives of Chitinophagaceae: 0.273% (+Au), 11.19% (+Ag), 21.5% (+Pb), 23.26% (+Ni), 25% (+Cr) and 26.38% (+Cd). The other unclassified microorganisms belonged to Porphyromonadaceae, Sphingobacteriaceae, Ohtaekwangia, Flavobacteriaceae, Prolixibacteraceae, Cytophagaceae and Chryseolinea, (S7 Table).

In the case of Firmicutes, 5 dominant genera (abundance >10%) were distinguished. 22 other, less abundant genera, listed in S8 Table, were grouped into one cluster named 'Other'. The reason for this was again the low proportion of Firmicutes (up to 0.87%) and the low abundance of 'Other' genera (0.27%) in the whole microbiome. The share of dominants is

presented in Fig 5. *Bacillus* was the most abundant genus of Firmicutes, its proportion was highest in +Ag (75.76%) followed by +Au (50.9%), +Pb (41.36%), +Cd (38.16%) and +Ni (36.67%). The lowest percentage of *Bacillus* sp. was found in NTP and Cr-treated microbiomes (7.3 and 14.94%, respectively). The second dominant was *Staphylococcus* sp., present in high numbers in all samples except silver-treatment (7.07%). The highest abundance was in NTP (58.39%) and +Ni (33.3%). Other treatments were characterized by abundances of 27% (+Cr), 27.96% (+Cd), 24.95% (+Pb) and 24.53% (+Au). The remaining 3 genera had a share of 10 percent or less. *Enterococcus* sp. was detected in +Cr (0.9%) and +Ag (10.1%). *Streptococcus* sp. was the most abundant in Au- (1.3%) and control (10.2%) treatments whilst other treatments had an abundance of <9% and there were no Streptococcus sp. in the Ag-treated microbiome. *Clostridium* sp. showed a 16.5% share in chromium-treatment, low abundance in Pb, Cr and Ag (<7%) and no presence in +Ni and +Au. The share of the 'Other' cluster was highest in Cd-, NTP and Cr-treatments, respectively: 24.67, 18.25 and 10.07%. The remaining treatments showed a share of <10%. Unclassified Firmicutes belonged to 3 families: Ruminococcaceae, Lachnospiraceae and Veillonellaceae. The latter constituted the highest proportion (33–100%) in unclassified microorganisms– 16.67% in +Ni, 14.95% in +Cr and 12.69% in +Pb (S7 Table). There were no unclassified Firmicutes in silver-treatment (Fig 5).

The non-exposed microbiome had only one representative of Actinobacteria, *Propionibacterium* sp. After exposition of *A. filiculoides* to the metals studied, *Propionibacterium* sp. declined (+Pb 22%, +Au 17.5%, +Ni 13.6%, +Cd 8.5%, +Cr 5.5%, +Ag 0%) and other 5 genera became dominant (Fig 6). Moreover, the presence of 26 less abundant genera (up to 0.45% of total microbiome) was recorded (up to 35%) and a noticeable proportion of unidentified microorganisms (up to 52%). *Kocuria* sp. was the second dominant genus present in the all metal-treated samples, except for +Ni. The highest abundance of *Kocuria* sp. was noted in +Ag (54.8%) followed by +Cd and +Pb (22.19% and 15.35%, respectively), it was further present in +Au (9.65%) and +Cr (6.66%). *Mycobacterium* sp. was the third dominant being the most abundant in Au- (28.95%), silver- (19.36%) and Cd-treated microbiomes (13.46%) with a low proportion in +Pb (4.56%) and +Cr (1.04%). The presence of *Frondihabitans* sp. was noted only in Cr- (51.48%) and Au-treatments (14.9%). *Micrococcus* sp. had the highest abundance in +Ni (22.7%) followed by +Cd (7.98%), +Pb (4.18%) and +Cr (4.1%). The last dominant was *Corynebacterium* sp., present in all metal-treated samples except of Ag. The highest percentage of this genus was recorded in +Ni (11.36%), followed by +Pb (9.5%), +Cd (7.2%), +Au (4.4%) and +Cr (2.8%). The cluster of 'Other' genera was the most abundant in Pb, Cd and Cr treated *A. filiculoides* (34.6, 30.7 and 26%, respectively). In the presence of noble metals, the proportion of these genera was 10.5% (+Au) and 19.3% (+Ag), while there were none present in nickel treatment. A detailed composition of this cluster is presented in S9 Table. The share of unclassified microorganisms was the highest in +Ni (52%) and +Au (14%) treatments. Remaining combinations showed them at a level below 10% (the lowest in +Cr) (Fig 6). Within 9 families, containing unidentified microbes, only representatives of *Microbacteriaceae* had the most visible share in the group of unrecognized bacteria (52% in +Ni) (S7 Table).

## Other – rare phyla

A detailed composition of low-abundance phyla is presented in Fig 7. In NTP, only Fusobacteria, represented by *Fusobacterium* sp., were observed. The Ag-treated microbiome displayed only the *Truepera* genus (Deinococcus-Thermus). The composition of the other samples, with the exception of +Au, which contained no rare phyla, was more complex. We recorded some unidentified genera belonging to Parachlamydiaceae (Chlamydiae), Anaerolineaceae (Chloroflexi), Leptotrichiaceae (Fusobacteria), Planctomycetaceae (Planctomycetes) and

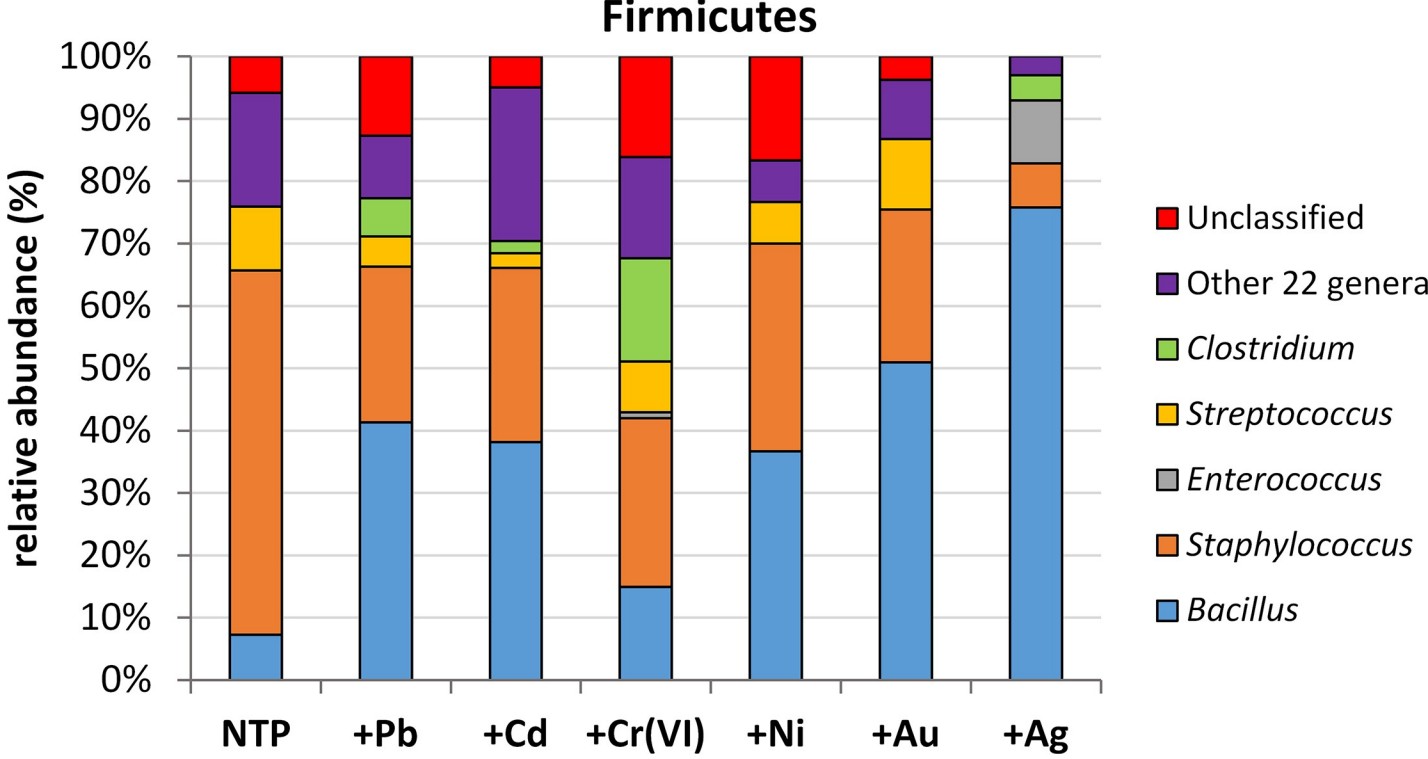

**Fig 5. The main genera dominants within firmicutes.** The group 'Other' represents those with abundance <10%.

Spirochaetaceae (Spirochaetes). Their abundance was highest (47.4%) in +Ni, followed by +Pb (25.4%), +Cd (25%) and +Cr (15.6%). Lead-treated microbiome has the following composition of the rare genera: 39.7% *Gemmatimonas* (Gemmatimonadetes), 12.7% *Opitutus*, 17.5% *Luteolibacter* (both Verrucomicrobia) and 4.8% Tepidisphaera (Planctomycetes). *Chthonomonas/ Armatimonadetes*_gp3 (Armatimonadetes) and *Pirellula* (Planctomycetes) were two main components of +Cd treatment constituting, 45 and 30%, respectively. Cr-treated *A. filiculoides* has the most diverse microbiome in rare phyla– 6 representatives were recorded: *Gemmatimonas* (33%), *Opitutus* (23%), *Nitrospira* (Nitrospirae, 11%), *Pelolinea* (Chloroflexi, 8.3%), *Caldisericum* (Caldiserica, 6.4%) and *Fusobacterium* (2.8%). The Ni-treated fern displayed only two of the above-mentioned genera of Verrucomicrobia with an equal proportion of 26.3% (Fig 7).

## Diversity and richness of the microbiome

Although the samples studied differed in a number of observed species (Sobs), we noticed two distinct groups: one of low richness 36–44 and the other with Sobs around 100 (Table 1). Non-treated *A. filiculoides* had one of the lowest observed species richness– 36. Similar numbers were recorded for treatments with noble metals. When *A. filiculoides* was exposed to heavy metals Sobs increased strongly for +Pb and +Cd (95 and 98) and slightly for +Ni (44). In the case of Cr-treatment, there was an almost 3 times higher number of species recorded (105) than in the control sample (Table 1). According to these data, the samples were divided into two groups: A–observed number of taxa similar to the control sample (36–44), B–representing the samples with a much higher number of observed taxa compared to the control sample (95– 105). We performed t-test on the groups mentioned and found that low number taxa group (A), including NTP, differed significantly from those of high richness (B, p<0.001).

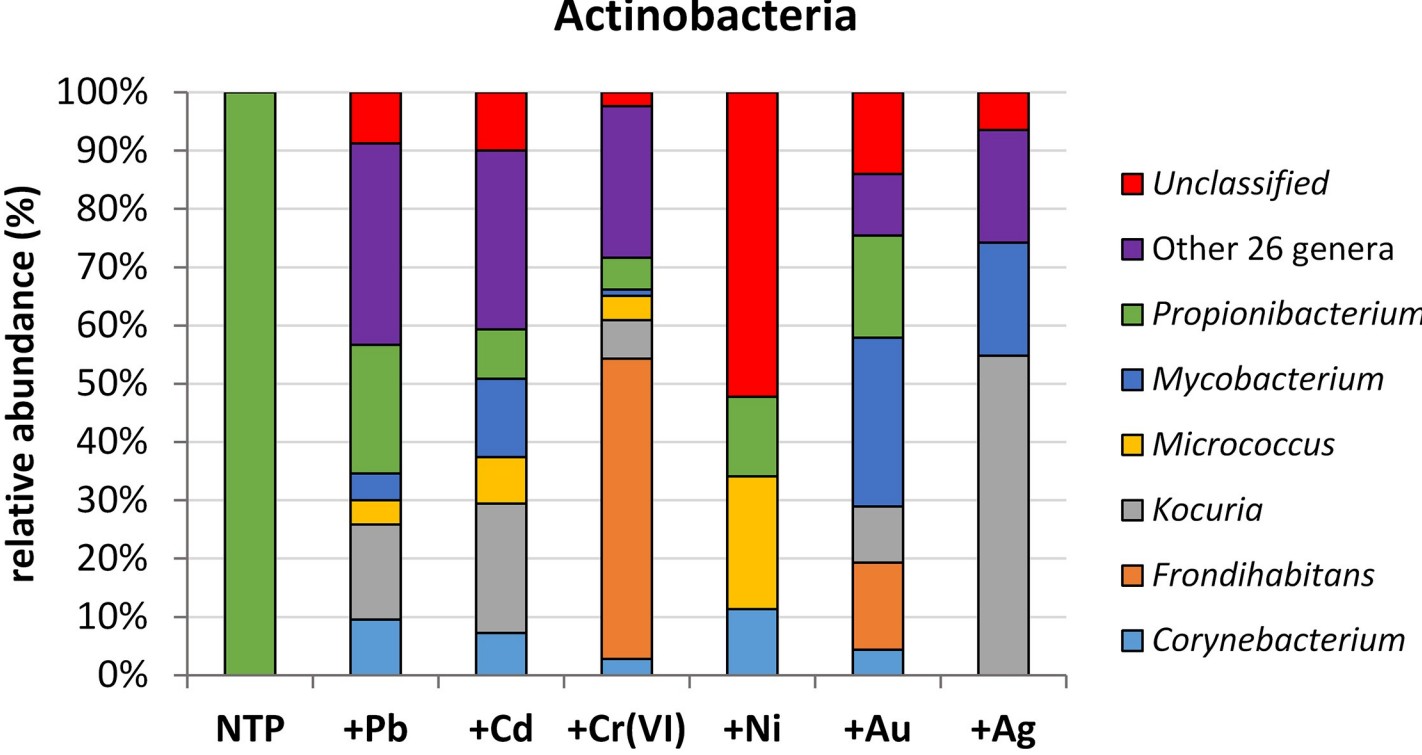

**Fig 6. The main genera dominants within actinobacteria.** Group 'Other' represents those with abundance <10%.

Generally, the biodiversity (*H'*) index reached higher values for metal-treated samples in relation to NTP. The highest values were noted for lead treatment followed by similarly biodiverse Ni and Au samples. This was followed by Cr and Ag treatments, which had similar biodiversity while the cadmium treated microbiome displayed the least diverse microbiome (Table 1). Samples +Pb, +Ni and +Au were characterized by the highest values of Simpson index (D) (higher number of dominants) followed by +Cr and +Ag. The lowest D were recorded for +Cd and control. Values for inverse Simpson indicated a higher diversity in all metal treatments except cadmium, in which it was even lower than in NTP. This pattern was the same with regard to richness values.

Interestingly, the numbers of observed taxa were similar in the control sample versus +Ni, +Au and +Ag treated (A-group) samples. However, the latter ones were characterized by significantly higher Shannon and Simpson indices, and a much lower evenness index. This phenomenon is related to the higher dominance of fewer taxa in the control sample in which only 4 taxa (*A. azollae*– 49%, *Brevundimonas* sp.– 32%, *Rhizobium* sp.– 14% and *Asticcacaulis* sp.– 4%) represent almost 99% of all microorganisms. Conversely, in the metal treated (A-group) there was a higher abundance of more taxa, for example, +Ni (8 taxa ~ 99%), +Au (8 taxa ~99%), +Ag (9 taxa ~99%). The highest number of observed taxa in the biodiversity group "B", was mostly related to low-abundant bacteria, for example only 7 taxa in +Pb treated samples were above 1%. For +Cd treated samples, this number is only 4, and 5 for +Cr(VI).

## Heavy metals and bacterial dominants

Principal Component Analysis (PCA) was applied to test the correlation of the metal treatment on microbiome composition. The results are presented in Fig 8.

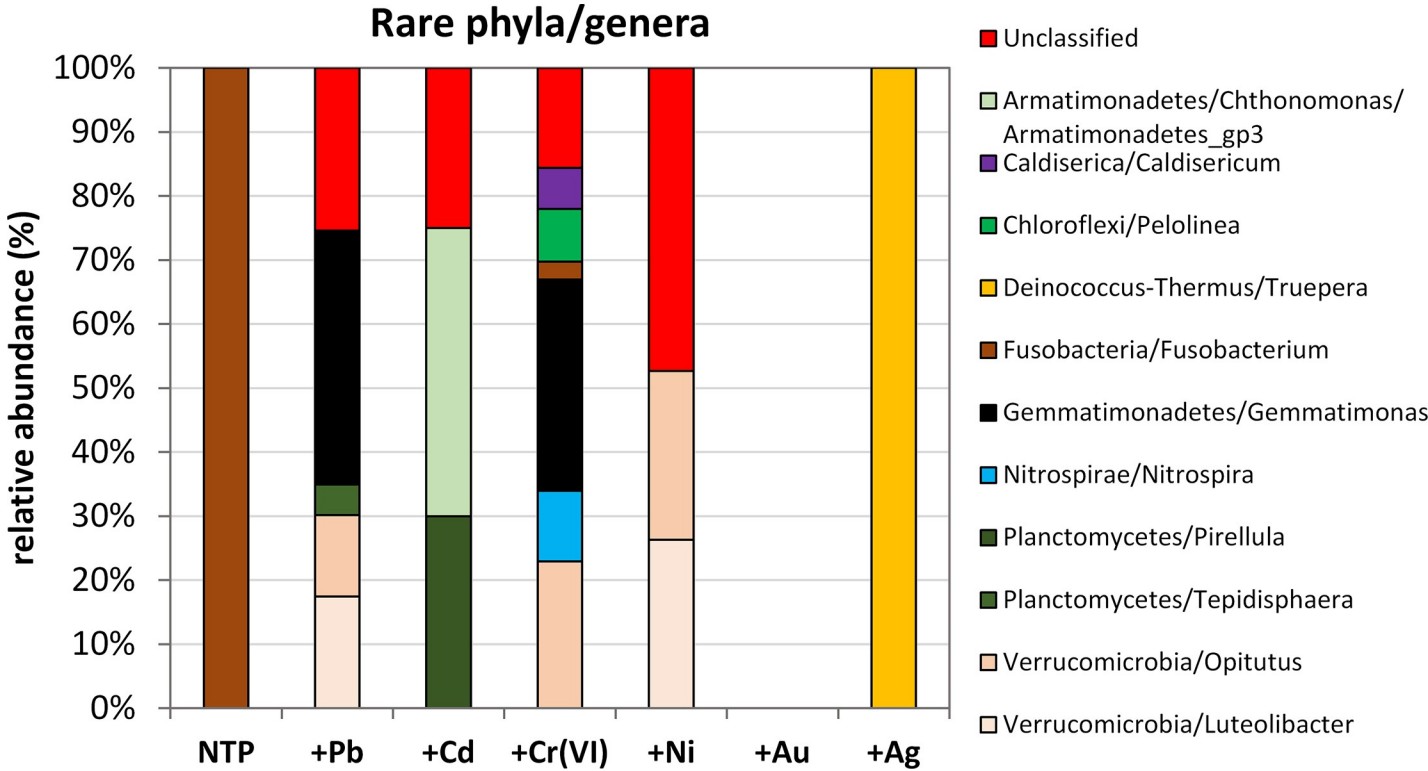

**Fig 7. The composition of rare phyla and corresponding genera of the 'Other' cluster.**

It was noticed that in non-treated *A. filiculoides* the dominant bacteria was *Brevundimonas* sp. However, after exposition to the metals studied, other specific genera become dominant. For Ni, *Asticcacaulis* sp. was the main dominant, and *Dyella* sp. in Au-treatment. The highest abundance of *Rhizobium* sp. in the Cd-treated sample was not surprising, as this genus is equipped with very effective molecular efflux mechanisms and a high tolerance to cadmium has been repeatedly proven [40,41]. There is hardly any noticeable effect of Cr(VI) and silver on the dominance of any specific group of bacteria.

## Discussion

The presented data provide a characterization of *Azolla filiculoides* endosphere bacterial communities under heavy metal stress. Obtained results demonstrated that overall richness (Table 1) was significantly higher for Pb, Cd and Cr-treated *A. filiculoides* than non-treated

**Table 1. Diversity and richness indices for each sample of studied microbiome.**

| Biodiversity group | Sample | $S_{obs}$ | Shannon (*H'*) | Simpson (*D*) | Inverse Simpson (*1/D*) | Evenness (*E*) |
|---|---|---|---|---|---|---|
| A | NTP | 36 | 0.974 | 0.523 | 2.098 | 0.626 |
| | +Ni | 44 | 1.778 | 0.795 | 4.870 | 1.082 |
| | +Au | 40 | 1.780 | 0.795 | 4.868 | 1.111 |
| | +Ag | 36 | 1.484 | 0.696 | 3.290 | 0.954 |
| B | +Pb | 95 | 1.967 | 0.793 | 4.833 | 0.995 |
| | +Cd | 98 | 1.220 | 0.437 | 1.776 | 0.612 |
| | +Cr(VI) | 105 | 1.511 | 0.679 | 3.114 | 0.748 |

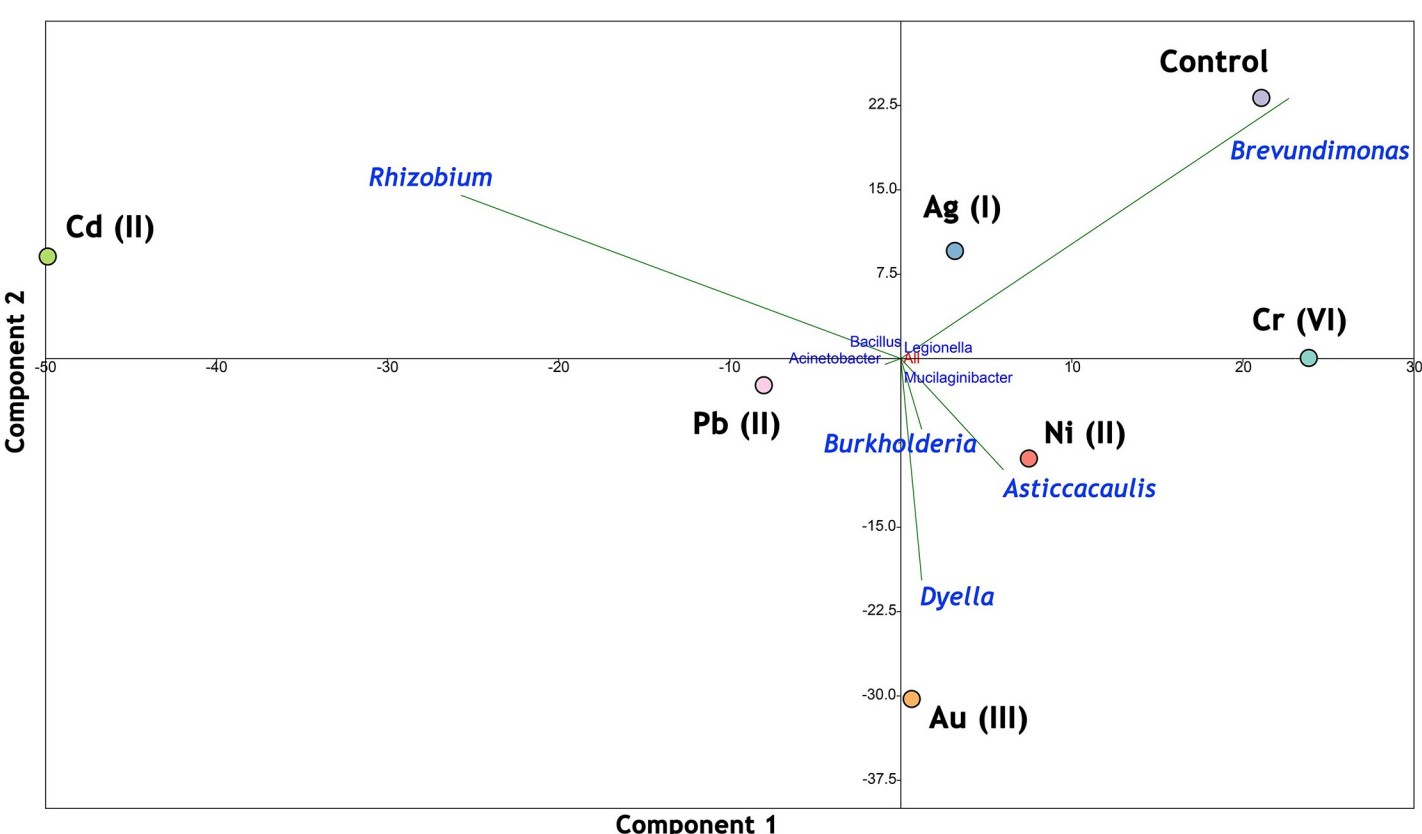

**Fig 8. Results of PCA for studied samples.**

and Ni, Au and Ag treatments. Only Pb, Cd and Cr resulted in higher abundances of taxa, which can be seen in Figs 1, 3, 4, 6 and 7. These results are not surprising, since previous studies have reported disturbances in rhizosphere communities due to metal stress [42–45]. On the other hand, Zedel and co-workers [45] demonstrated that overall richness and bacterial 16S rRNA copy numbers did not differ between control and heavy metal treatment.

Our study allowed us to detect metal-resistant microorganisms observed in all samples constituting core microbiome of *A. filiculoides* which may be important in metal removal. We constructed a table (Table 2) showing a list of the most resistant genera ranging from the most to the least 10 genera of core microorganisms belonging to: *Acinetobacter*, *Asticcacaulis*, *Anabaena*, *Bacillus*, *Brevundimonas*, *Burkholderia*, *Dyella*, *Methyloversatilis*, *Rhizobium* and *Staphylococcus* were determined.

The SUM column is the number of treatments in which a given genus was observed. Microorganisms making the core microbiome are stated in the CORE column (in grey). In addition, the superscript letter T indicates microorganisms known as metal tolerant according to the literature.

## Core microbiome

*Anabaena azollae* was the most abundant microorganism in the core microbiome, constituting 36–54% of the total microbiome and present in all samples. In Surosz and Palinska's study [46], it was demonstrated that both Cd and Cu had a negative impact on free-living *A. flos-aquae*, decreasing its growth down to 50%. We evidenced that the abundance of endophytic

**Table 2. List of the proposed genera resistant to the metals studied and organized according to their abundance in the total microbiome in descending order.**

| Genus | NTP | +Pb | +Cd | +Cr(VI) | +Ni | +Au | +Ag | SUM | CORE |
|---|---|---|---|---|---|---|---|---|---|
| *Anabaena* | X | X | X | X | X | X | X | 7 | Y |
| *Rhizobium* | X | X | X | X | X | X | X | 7 | Y |
| *Brevundimonas* | X | X | X | X | X | X | X | 7 | Y |
| *Asticcacaulis* | X | X | X | X | X | X | X | 7 | Y |
| *Dyella* | X | X | X | X | X | X | X | 7 | Y |
| *Burkholderia* | X | X | X | X | X | X | X | 7 | Y |
| *Acinetobacter*[T] | X | X | X | X | X | X | X | 7 | Y |
| *Methyloversatilis* | X | X | X | X | X | X | X | 7 | Y |
| *Staphylococcus*[T] | X | X | X | X | X | X | X | 7 | Y |
| *Bacillus*[T] | X | X | X | X | X | X | X | 7 | Y |
| *Acidisoma* | | X | X | X | X | X | X | 6 | |
| *Sediminibacterium* | | X | X | X | X | X | X | 6 | |
| *Sphingomonas* | | X | X | X | X | X | X | 6 | |
| *Streptococcus* | X | X | X | X | X | X | | 6 | |
| *Propionibacterium* | X | X | X | X | X | X | | 6 | |
| *Mucilaginibacter*[T] | | | X | X | X | X | X | 5 | |
| *Pseudomonas*[T] | X | X | | X | X | X | | 5 | |
| *Pelomonas* | | X | X | | X | X | X | 5 | |
| *Mycobacterium*[T] | | X | X | X | | X | X | 5 | |
| *Corynebacterium*[T] | | X | X | X | X | X | | 5 | |
| *Nocardioides* | | X | X | X | | X | X | 5 | |
| *Stenotrophomonas*[T] | X | X | X | | X | | | 4 | |
| *Clostridium_sensu_stricto*[T] | | X | X | X | | | X | 4 | |
| *Micrococcus*[T] | | X | X | X | X | | | 4 | |
| *Achromobacter*[T] | X | X | X | | X | | | 4 | |
| *Geobacter*[T] | X | X | | X | X | | | 4 | |
| *Flavobacterium*[T] | X | X | | X | | | | 3 | |
| *Arthrobacter*[T] | | X | X | | | | | 2 | |
| *Delftia*[T] | X | | | | | | | 1 | |
| *Clostridium_XlVa*[T] | | X | | | | | | 1 | |

living *A. azollae* was lower by up to 13%. However, this phenomenon was determined only in respect to Ni, Cr and Au treatments. This fact could be explained by negative effects of metals on the levels of chlorophylls, the efficiency of the photosynthetic apparatus, and the protein contents [46,47]. Cyanobacteria responded by producing polyphosphate granules to immobilize Cu. It should be mentioned that heavy metals could be adsorbed and transported into cells of cyanobacteria and eventually stored in organelles [48]. This, on the one hand may negatively affect the ultrastructure of cyanobacteria but, on the other hand, could help with metal removal as well as induce the generation of reactive oxygen species (ROS). Cyanobacteria possess two defense systems: enzymatic antioxidant defensive system (superoxide dismutase, catalase, ascorbate peroxidase) and non-enzymatic compounds consisting of ascorbic acid, reduced glutathione, carotenoid and proline [48]. This knowledge can explain high abundance of *A. azollae* and the tolerance for particular metals (Pb, Ag and Cd).

*Rhizobium*, the second most abundant genus (8.78–34.47%), is known to be resistant to Cd by using several mechanisms such as: extracellular immobilization, periplasmic allocation, cytoplasmic sequestration, and biotransformation of toxic products. It has been shown that increased levels of glutathione were key factors in *Rhizobium* sp. tolerance to Cd [40,41]. This

feature could explain the highest abundance of *Rhizobium* sp. in Cd-treated *A. filiculoides* (34.5%).

The potential of *Brevundimonas* spp. to remove Co and Ni was previously demonstrated by Fosso-Kankeu and co-workers [49] via absorption into their cell walls, which was hampered in the presence of lighter ions such as $Mg^{2+}$ and $Ca^{2+}$. A study on the influence of arsenic on *Brevundimonas diminuta* NBRI012 isolated from rice (*Oryza sativa* L. Var. Sarju 52) revealed the ability of bacterium to promote plant growth as well as accumulation of As in the biomass, which has a positive effect on rice growth [50]. In our study, the bacterial genus mentioned constitutes 3.86–32.4% of the total microbiome present in the endosphere of *A. filiculoides* showing important role of this genus in heavy metal accumulation.

*Asticcacaulis* sp. was another abundant bacterial strain (3–19%) in the core microbiome. However, there is a lack of sufficient information on its tolerance to heavy metals. The study of Luo et al. [51] evidenced that endospheric *Asticcacaulis biprosthecium* together with members of endospheric Sphingomonadaceae i.s. and Pseudomonadaceae i.s. may affect *Sedum alfredii* Hance shoot Zn content (Pearson's r 0.965*) or Cd hyperaccumulation (Pearson's r 0.996**). This makes *Asticcacaulis* sp. beneficial microorganisms supporting Zn and Cd accumulation by *Azolla* sp. Babich and Stotzky [52] demonstrated that the growth of *Asticcacaulis excentricus* was inhibited at 10 ppm of Ni and completely inhibited at 40–50 ppm of Ni. There were no reports on mechanisms providing for the survival of *Asticcacaulis* sp. under exposition to heavy metals. In our study we found this bacterium constituting 3% (Cd-treatment) and 7% (+Ni) of the microbiome.

*Dyella* sp. is a microorganism involved in the degradation processes, especially associated with the biodegradation of aromatic compounds [53], azo dyes and heavy metals (Cu and Ni) [54]. However, there is a lack of knowledge about the potential of *Dyella* sp. in heavy metal remediation. In our experiment, *Dyella* sp. abundance oscillated in a range from 0.05 to 17.85% constituting noticeable fraction of microorganisms for almost all metal treatments except lead what may show its important role in the removal of these metals.

Liu et al. [55] isolated *Burkholderia fungorum* FM-2 from oil-contaminated soil and proved that this strain utilized both phenanthrene as well as Pb(II), Cd(II) and Zn(II). Based on genomic, physiological and biochemical analyses, Wang and co-workers [56] demonstrated that *Burkholderia cenocepacia* YG-3 is well-adapted to Cd removal and has the potential to cooperate with its host to improve phytoremediation efficiency in heavy metal contaminated sites. These authors suggested that YG-3 strains possess a complex mechanism to adapt to Cd stress: (1) binding Cd to prevent it from entering the cell via cell wall components; (2) producing siderophores and exopolysaccharides for Cd immobilization; (3) intracellular sequestration of Cd by metalloproteins; (4) excretion of Cd from the cell by efflux pumps; (5) alleviation of Cd toxicity by antioxidants [55]. Recent literature indicates that microorganisms belonging to *Massilia* and *Burkholderia* genera were mercury-tolerant [56]. The abundance of these bacterial genera in *A. filiculoides* microbiome ranged between 0.03 and 8.79%, specifically 5.56% and 0.4% in Pb- and Cd-treated plant, respectively.

In our studies, bacterial representatives belonging to *Acinetobacter* genus seemed to be in seventh position according to their abundance (0.07–4.9%). Šipošová et al. [57] showed that genetic variability in *Acinetobacter* sp. (isolated from soil and drainage water samples collected from three areas in Slovakia that had been industrially polluted with heavy metals) is high and what is more in different species various numbers of genetic orthologs could be observed. These authors suggested that the presence of plasmids is one of the sources of genetic diversity and that frequently, heavy metal resistance in acinetobacters is plasmid encoded [57]. Other researchers also showed multiple plasmids (plasmidome) encoding resistance to Hg, Cr, Co/Zn/Cd, Ni, and As in *Acinetobcter* strains and suggested that these plasmidomes can provide

resistance to heavy metals in both environmental and clinical habitats [58–60]. El-Sayed [61] determined dual expressions of heavy metal and antibiotic resistance from *A. baumannii* strain HAF– 13, a potential seed for decommissioning of sites polluted with industrial effluents rich in heavy metals, since this isolate will be able to withstand in situ antibiosis that may prevail in such ecosystems. In another study, Furlan et al. [62] presented that *A. seifertii* strain SAb133 have the highest tolerance to metals. In our previous studies [21], *Acinetobacter* endophytic strain was found as a good source of extremophilic bacteria widely used in modern biotechnology. Described mechanisms may be responsible for tolerance of studied metal by *Acinetobacter* sp. whose abundance was highest for lead-treated plant. In our previous study we demonstrated that *Acinetobacter* sp. AzoEndo8 synthetized 3 out of 6 growth promoting substances what makes it moderately useful in the supporting growth of *Azolla* sp. [21].

In the work of Fernandes and co-workers [63], the resistance of *Methyloversatilis* sp. to Cd (40 ppm), Hg (5 ppm) and Pb (500 ppm) was studied, and they proved that this bacterial genus is also able to remove antimony(III) from soils. In the current study, rather low abundance of *Methyloversatilis* sp. (up to 0.82% in +Cd) was found.

*Staphylococcus* was one before last genus forming the core microbiome of *A. filiculoides* with an abundance of 0.37%. Its tolerance to heavy metals (Ni, Cr(VI) and Cd) was earlier demonstrated [64] and is often connected with a resistance to antibiotics [64,65]. Moreover, the sensitivity of *S. xylosus*, *S. aureus*, *S. capitis*, *S. lentus*, *S. epidermidis*, *S. sciuri* and *S. chromogenes* to silver nitrate was revealed [66]. It was also reported that the process of particular metal removal (Pb, Cd, As, Cr, Hg) was based on biosorption phenomena [66]. We found the highest abundance of this genus for Pb, Cd and Cr suggesting important role of *Staphylococcus* sp. in metal removal. In addition we showed that *Staphylococcus* sp. AzoEndo11 was able to synthetize 4 out of 6 growth promoting substances which make it moderately-good microorganism in supporting *Azolla* sp. [21].

*Bacillus*, a very popular genus, is commonly associated with plants and supports their growth [1]. Examples of associations with hyperaccumulator plants are: *Thlaspi goesingense*, *Alyssum bertolonii* [67], *Alnus firma* [68], *Azolla filiculoides* [21]), *Solanum nigrum* [69], *Brassica juncea* [70], *B. napus*, *Nocceae caerulescens*, *Pteris multifidia*, *Petris vittata*, *Solanum nigrum* [71], *Elsholtzia splendens* [72]. In our study, *Bacillus* sp. was present in all samples and its abundance was higher under exposition to heavy metals (up to 27 times in case of Pb-treatment). This is congruent with our previous study [21], where 46.55% of identified isolates belonged to the genus *Bacillus*. In addition, its potential in the promotion of *A. filiculoides* growth was demonstrated [21]. Mondal and co-workers [73] studied the potential of *Bacillus* sp. KUJM2 for remediation of potentially toxic elements. The mechanisms of metal removal were connected with their binding to the bacterial cell wall due to the presence of functional groups such as carboxyl groups, which under higher pH become deprotonated and allows for metal binding [73]. *Bacillus* sp. can also immobilize metals (e.g. Cd, Pb, Cu, Ni, Zn, As) [68] in soil by producing polymeric substances which can effectively chelate metal ions [74]. In that way, the harmful effects of metals on plants are lowered by reduced metal bioavailability, uptake and translocation [73] and allows for enhanced heavy metal phytoremediation [68]. In our previous study [21] the potential of two *Bacillus* strains in producing substances promoting plant growth was demonstrated. Endophytic *Bacillus* sp. AzoEndo3 produced 5 out of 6 compounds whilst epiphyte, *Bacillus* sp. AzoEpi2 only three of them.

## Other possible metal tolerant strains

Besides the 10 top strains that make up the core microbiome described above we also detected microorganisms not present in all treatments; some of them were not detected in non-treated

*A. filiculoides* (Table 2). They attracted our attention even though their abundances were mostly less than 0.1%. Higher abundances were noted only for the following genera: *Sedimini-bacterium* (0.62%), *Sphingomonas* (0.26%), *Mucilaginibacter* (1.69%), Pseudomonas (0.79%), *Pelomonas* (0.2%), *Stenotrophomonas* (0.15%) and *Clostridium_sensu_stricto* (0.14%). Interest-ingly, some of them have been reported as metal tolerant.

*Mucilaginibacter* sp. is known to be metal-resistant bacteria. In the study of Li and co-work-ers [66], two strains, *M. rubeus* and *M. kameinonensis*, were isolated from a gold/copper mine in China. The authors showed bacterial tolerance to Cu, Zn, Cd and As(III) and identified genes that encode the proteins responsible for the metal resistance such as: putative $P_{IB-1}$-ATPase, putative $P_{IB-3}$-ATPase, putative Zn(II)/Cd(II) $P_{IB-4}$ type ATPase, and putative resis-tance-nodulation-division (RND)-type metal transporter systems [75].

Other authors [76] reported the potential of various strains of *Pseudomonas* sp. to tolerate Pb, Zn, Cd, Cu, Cr and Ag. These strains possessed plasmids and metal-resistance genes such as NMGD, *pbr*A and *chr*A. Moreover, in this study the same genes were reported for *Bacillus* sp. The potential of *Pseudomonas putida* to tolerate Ni (up to 40–50 ppm) was reported by Babich and Stotzky [52]. Tolerance of *Pseudomonas aeruginosa* to Cu, Cd, Zn and Co was pre-sented by Mihdhir and co-workers [77].

Wood and co-workers [78] presented a list of microorganisms tolerant to given metals: sev-eral *Pseudomonas* strains (Cd, Pb, Ni), *Arthrobacter mysorens* 7 (Cd, Pb), *Mycobacterium* sp. ACC14 (Cd), *Achromobacter xylosoxidans* Ax10 (Ni) and *Delftia* sp. (As). *Corynebacterium glutamicum* was reported to be one of the most arsenic-resistant microorganisms [79]. The analysis of its genome revealed the presence of two complete ars operons (*ars1* and *ars2*) com-prising the typical three-gene structure *arsRBC*, with an extra *arsC1'* located downstream from *arsC1* (*ars1* operon), and two orphan genes (*arsB3* and *arsC4*) [79]. *Stenotrophomonas* sp. was successfully applied for absorption of Au(III) [80]. In 1994, *Clostridium* sp. was shown to be able reduce U(VI) and stabilize uranium waste [81]. The presence of uranium tolerant species makes the microbiome (and whole plant) also very attractive in purification of aquifers with radionuclides as demonstrated recently by Xinwei and co-workers [23] or earlier by Sela and Tel-Or [24].

Similarly to *Clostridium* sp., *Geobacter* sp. is also involved in multiple metal reduction pro-cesses including Cr(VI), U(VI), and Fe(III) and perform in situ bioremediation at high levels of Cr contamination [82]. The potential of *Micrococcus* sp. to remediate Cr and Cu was pre-sented by Oyewole and co-workers [83] while *Flavobacterium* sp. was able to tolerate Cu at 100 mg $L^{-1}$ [84].

These microorganisms were interesting for increasing the survival of *Azolla* sp. under heavy metal stress. In our previous study [21] we found *Delftia* sp. AzoEpi7 as the most effi-cient microorganism in promoting growth of *A. filiculoides*. *Achromobacter* sp. AzoEpi1 and *Micrococcus* sp. AzoEndo14 showed partial role in synthetizing 4 and 5 out of 6 plant growth promotors, respectively [21].

The presence of the unique internal plant microbiome influences the metal uptake of plant. Epiphytes could immobilize metals and make them unavailable for plants while endophytes could capture them within plant tissues to prevent their effect on plant metabolism. Moreover, plant tissues and intercellular spaces are much more optimal habitat for endophytes than soil. For that reason, it could be stated that endophytes possess higher performance than soil micro-organisms [45]. The localization of metals in plant organs depends on their mobility. In the study of Sela and Tel-Or [24], copper and uranium were accumulated mainly in roots whilst cadmium in both shoots and roots to a similar level. Cadmium formed precipitates with phos-phate and calcium in xylem cells of the shoot bundle, uranium immobilization was also con-nected with calcium presence. In another study, these authors determined the accumulation of

Cd in the inner epidermis, cortex and bundle cell walls of the root [85]. In case of lead it precipitated in the vacuoles of mesophyll cells of *Azolla* plants [25]. Lead may also affect plant morphology by inhibiting the cell division or negatively influence on membranes integration. It may also disturb water status and lower nutrient uptake. The latter may affect photosynthesis rate which can be also decreased by decomposition of chlorophyll due to lead presence what was demonstrated for *Brassica napus* [86]. Similarly in other study Ali and co-authors [87] showed that cadmium also damaged cell structures, i.e. membranes and chloroplasts of the same plant. In another study [88] the toxic effects of chromium on *B. napus* were demonstrated–plant growth was hampered due to hindering the nutrients availability to aerial plant parts. In addition, the photosynthetic apparatus was damaged resulting in lower biomass.

It could be noticed that most of recognized metal-tolerant bacteria showed potential for the accumulation of Pb, Cd and Cr. It could mean that they possess mechanisms to tolerate these metals the most and this could be the reason that the highest abundances were noted in Pb-, Cd- and Cr-treatments.

Reports about sufficient experimental evidence has proven that plants that grow in unfavorable environments such as salty, dry, hot and heavy metal rich environments may develop different adaptation capacities to adversity and this may be partially due to symbiotic microorganisms [89]. There is a lot of scientific evidence about the possibilities of metal utilization by *Azolla* sp. [22]. Zadel and co-workers [45] concluded that metal tolerant plants might be selecting for a mainly metal-resistant/tolerant microbiome. In our opinion, this conclusion has very probably been confirmed by the presented study of the *Azolla filiculoides* microbiome. It was also evidenced that endophytes in hyperaccumulator plants were found to be more metal-resistant than its rhizosphere isolates [90]. Therefore, we suggested *Azolla filiculoides* as potential reservoirs for isolating bacteria effective at alleviating heavy metal stress in the plant, for reducing the accumulation of heavy metals in plants, for example crops, and removing heavy metals in aqueous media (bioremediation of heavy metal-contaminated wastewater system).

## Conclusions

This study revealed the presence of the microbiome of the aquatic fern *Azolla filiculoides*, whose diversity was strongly affected after exposition of the plant to the metals studied. In general, diversity of the microbiome increased in comparison to the non-treated plant. Taking into account the richness of the observed taxa, diversity of the microbiome was significantly higher only for Pb, Cd and Cr-treatments in comparison to the non-treated sample. We were able to identify the 10 top genera present in all samples, which constituted the core microbiome. Some of which are known to be metal tolerant. What more, additional genera of noticeable abundance in metal-treated *A. filiculoides* were recorded which are also known to be metal tolerant.

In summary, we have shown that *A. filiculoides* possess a microbiome whose representatives belong to metal-resistant species and which makes the fern a source of biotechnologically useful microorganisms for remediation process. In the future, it will be interesting to further study detected metal-tolerant microorganisms to know their potential in both metal removal and promoting *Azolla* sp. growth under exposition to heavy metals. In addition, the identification genes and mechanisms involved in metal tolerance by the bacteria will provide important knowledge useful for their future application in metal bioremediation not only in the co-operation with *Azolla* sp. but also to support metal phytoremediation by other aquatic plants.

## Supporting information

**S1 Table. The composition of 'Other' cluster of the representatives of alphaproteobacteria (percentage of whole proteobacteria).**
(DOCX)

**S2 Table. The composition of 'Other' cluster of the representatives of betaproteobacteria (percentage of whole proteobacteria).**
(DOCX)

**S3 Table. The composition of 'Other' cluster of the representatives of deltaproteobacteria (percentage of whole proteobacteria).**
(DOCX)

**S4 Table. The composition of 'Other' cluster of the representatives of gammaproteobacteria (percentage of whole proteobacteria).**
(DOCX)

**S5 Table. The relative abundance (%) of unidentified microorganisms (percentage of a given class of proteobacteria) for each treatment and their assignment to corresponding family.**
(DOCX)

**S6 Table. The composition of 'Other' cluster for each treatment presented as relative abundance (%) of the phylum bacteroidetes.**
(DOCX)

**S7 Table. The relative abundance (%) of unidentified microorganisms (percentage of a given phylum) for each treatment and their assignment to corresponding family.**
(DOCX)

**S8 Table. The composition of 'Oher' cluster for each treatment presented as relative abundance (%) of the phylum firmicutes.**
(DOCX)

**S9 Table. The composition of 'Other' cluster for each treatment presented as relative abundance (%) of the phylum actinobacteria.**
(DOCX)

## Author Contributions

**Conceptualization:** Artur M. Banach, Agnieszka Kuźniar.

**Data curation:** Artur M. Banach, Jarosław Grządziel.

**Formal analysis:** Artur M. Banach, Jarosław Grządziel.

**Investigation:** Artur M. Banach, Agnieszka Kuźniar.

**Methodology:** Artur M. Banach, Agnieszka Kuźniar.

**Project administration:** Agnieszka Wolińska.

**Resources:** Artur M. Banach, Agnieszka Kuźniar, Jarosław Grządziel, Agnieszka Wolińska.

**Software:** Jarosław Grządziel.

**Validation:** Artur M. Banach, Agnieszka Kuźniar.

**Visualization:** Artur M. Banach, Jarosław Grządziel.

**Writing – original draft:** Artur M. Banach, Agnieszka Kuźniar, Jarosław Grządziel.

**Writing – review & editing:** Agnieszka Kuźniar, Jarosław Grządziel, Agnieszka Wolińska.

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
