## [Decision Letter · Decision Letter 0]

18 Feb 2020

PONE-D-20-02088

The endophytic microbiome of Azolla filiculoides L. exposed to selected heavy metals

PLOS ONE

Dear Dr. Artur Banach,

Thank you for submitting your manuscript to PLOS ONE. After careful consideration, we feel that it has merit but does not fully meet PLOS ONE’s publication criteria as it currently stands. Therefore, we invite you to submit a revised version of the manuscript that addresses the points raised during the review process.

I agree with the reviewers that main claims of the paper are not properly placed in the context of previous literature. Authors should indicate the basis dictating the selection of different metals retained for this study.

We would appreciate receiving your revised manuscript by 45 days. To enhance the reproducibility of your results, we recommend that if applicable you deposit your laboratory protocols in protocols.io, where a protocol can be assigned its own identifier (DOI) such that it can be cited independently in the future. For instructions see: http://journals.plos.org/plosone/s/submission-guidelines#loc-laboratory-protocols

We look forward to receiving your revised manuscript.

Kind regards,

Basharat Ali, Ph.D

Academic Editor

PLOS ONE

Additional Editor Comments (if provided):

I agree with the reviewers that main claims of the paper are not properly placed in the context of previous literature. An overhaul is required for the Introduction section. Authors should indicate the basis dictating the selection of different metals retained for this study.

Journal Requirements:

Please ensure that your manuscript meets PLOS ONE's style requirements, including those for file naming. The PLOS ONE style templates can be found at http://www.plosone.org/attachments/PLOSOne_formatting_sample_main_body.pdf and http://www.plosone.org/attachments/PLOSOne_formatting_sample_title_authors_affiliations.pdf

Reviewers' comments:

Reviewer's Responses to Questions

**Comments to the Author**

1. Is the manuscript technically sound, and do the data support the conclusions?

Reviewer #1: Yes

Reviewer #2: Yes

2. Has the statistical analysis been performed appropriately and rigorously? 

Reviewer #1: Yes

Reviewer #2: N/A

3. Have the authors made all data underlying the findings in their manuscript fully available?

Reviewer #1: Yes

Reviewer #2: Yes

4. Is the manuscript presented in an intelligible fashion and written in standard English?

Reviewer #1: Yes

Reviewer #2: No

5. Review Comments to the Author

Reviewer #1: The present study (The endophytic microbiome of Azolla filiculoides L. exposed to selected heavy metals) explores an exciting topic linking plants and microorganisms for a concerted role in sustainable management of metal-contaminated ecosystems. Insightful results have been obtained that advance knowledge towards the use of Azolla filiculoides in applied technologies for remediation of metal-contaminated environments.

In general, the manuscript is well written and appears well balanced, though a number of issues have to be addressed.

The title of the manuscript could be refined to clearly express either the objective of the study or the significance of obtained results.

The second part of the abstract has to be improved. Particularly, the comparative behavior of different strains under different metals treatments should be highlighted. Further, different results presented should be quantified (%, etc.).

An overhaul is required for the Introduction section. Following crucial information could be integrated:

* Please elaborate on the behavior of metals (particularly the ones selected for this study) in the environment and in biological systems, with reference to their physico-chemical properties. This would smooth the justification of using different metals in this study.

* Further, please incorporate available information on the responses of Azolla plants under metals stress conditions. Likewise, the responses of endophytic microbes to metal stress could be incorporated in a way supporting the hypothesis of the study.

* Is Azolla filiculoides a metal hyperaccumulator (L21)? Related information and reference can be incorporated in the Introduction section to support this statement.

* According to sentence in L74-76, authors seem to indicate a double objective for current study i.e (i) to assess the tolerance of Azolla microbiome, (ii) to assess their role in assisted phytoremediation. Has the second objective been investigated?

In the Materials and Methods section, authors should indicate the basis dictating the selection of different metals retained for this study.

L107: is it 3 or 2 g portion of the sample?

L102: Which part of the plant was sampled? Were plants materials collected to make bulked samples? Please precise. It is commonly believed that the abundance of endophytes varies with regard to plant part. If that is true for Azolla, can authors ascertain whether the procedure used for samples collection didn’t influence results on endophytes abundance?

Authors reported higher richness of endophytes under Pb, Cd and Cr-treatements than Ni, Au and Ag treatments. It is of great interest to discuss the possible (s) raison (s) behind such differential effects of metals.

Reviewer #2: General comments:

The present manuscript entitled " The endophytic microbiome of Azolla filiculoides L. exposed to selected heavy metals” contains a good idea and impressive amount of work to explain the tolerance ability of target fern (in combination with microbiome) exposed to selected heavy metals. This research is problem-based and might help scientists working on phytoremediation or bioremediation. Nevertheless, authors need some improvement in the manuscript for their possible acceptance in the Journal. Following are some suggestions to be incorporated before making any decision.

Specific Comments:

Abstract section:

Make clear hypothesis at the start of this section.

Overall it is fine

Introduction section:

In line 42-43: Rephrase the sentence. Also, delete the extra addition of “and” and which” from these lines.

In line 47-48: Delete extra commas (,)

In line 53-56: Rewrite the sentence to make it clearer. Avoid repetition of words

In line 72: “One plant” should be “Among aquatic ferns” or “Among hyperaccumulators” and “heavy metal phytoremediation” should be “heavy metal removal by phytoremediation”, change accordingly.

In line 75: Mention the particular heavy metals

In line 77-78: Clearly write the sentence i.e. “their………microorganisms”

It is suggested to add few sentences describing the phytotoxic impacts of selected heavy metals and their removal by phytoremediation strategies.

Further improve the study objectives at the end of this section.

Materials and methods section:

In line 98: The selection of particular heavy metals and their doses are not clearly described. Please explain them within the text or provide reference (not unpublished data).

In line 102: The collection of plant material was carried out after one week, specify the reason?

In line 149-151: Re-arrange the sentence

Results section:

In line 185 and so on: Please cite all figure numbers inside the text (not with headings).

In line 244: Delete either “percentage” or “%”.

In line 247: Use “%” with last number/value only.

In line 267: First use “lead (Pb)” and then “Pb” (also for other metals) throughout the section. Be consistent

In line 352: Correct as “―gp3”

Explanation of results is fine.

Discussion section:

In line 407: “A. filiculoides” should be Italic and write genus “Azolla” due to first appearance.

In line 410: “Only Pb, Cd and Cr resulted in higher abundances of taxa”, Please check “Ag” also showing significant abundance such as in figure 1.

In line 465-466: Re-write the sentence “In 1983…………microorganisms”

In line 507:” which” should be replaced with “whose” and “was the highest” should be “was highest”. Please re-check in other sections

Authors are suggested to further discuss this section with more possible mechanistic approaches. There are some related articles which can help the authors to understand the similar topic; Biologia Plantarum, 2014, 58(1), 131-138; Ecotoxicology and Environmental Safety, 2014, 102, 25-33; and Plant Physiology and Biochemistry, 2019, 145, 142-152.

Conclusion section:

At the end of this section, it is suggested to add one or two more sentences concerning the future prospects of the present study.

Overall, there are some grammatical errors throughout the manuscript. All sections of the present study were well enough explained and discussed (although need correction/improvement).

6. PLOS authors have the option to publish the peer review history of their article (what does this mean?). If published, this will include your full peer review and any attached files.

Reviewer #1: No

Reviewer #2: No

---

## [Author Response · Author response to Decision Letter 0]

6 Apr 2020

Dear Reviewers,

Please find our responses to your comments below. We have improved the manuscript and believe it will satisfy everybody.

For line number we used those from previous PDF file. As we added new references the list of them was re-number

With kind regards, on behalf of co-authors,

Artur Banach, PhD.

Reviewer #1:

The present study (The endophytic microbiome of Azolla filiculoides L. exposed to selected heavy metals) explores an exciting topic linking plants and microorganisms for a concerted role in sustainable management of metal contaminated ecosystems. Insightful results have been obtained that advance knowledge towards the use of Azolla filiculoides in applied technologies for remediation of metal-contaminated environments.

In general, the manuscript is well written and appears well balanced, though a number of issues have to be addressed.

Thank you very much for this encouraging statement, nowadays study on microbiomes are very popular and important for example for agriculture or pollutants removal. We would like to deepen this topic for Azolla which is very interesting plant called ‘superorganism” having many applications.

1. The title of the manuscript could be refined to clearly express either the objective of the study or the significance of obtained results.

We could agree to change the title into addressing more the objective/significance of our findings: “Azolla filiculoides L. as a source of metal-tolerant microorganisms”

2. The second part of the abstract has to be improved. Particularly, the comparative behavior of different strains under different metals treatments should be highlighted. Further, different results presented should be quantified (%, etc.).

We added more numerical information in the abstract:

We removed the sentence in line 33 and added from line 25: “The main dominants were Cyanobacteria and Proteobacteria constituting together more than 97% of all reads. Metal treatment led to changes in the composition of the microbiome and showed significantly higher richness in the Pb-, Cd- and Cr-treated plant in comparison with other (95-105 versus 36-44%). In these treatments the share of subdominant Actinobacteria (0.4-0.8%), Firmicutes (0.5-0.9%) and Bacteroidetes (0.2-0.9%) were higher than in non-treated plant (respectively: 0.02, 0.2 and 0.001%) and Ni-, Au- and Ag-treatments (respectively: <0.4%, <0.2% and up to 0.2%). The exception was Au-treatment displaying the abundance 1.86% of Bacteroidetes.”

3. An overhaul is required for the Introduction section. Following crucial information could be integrated:

• Please elaborate on the behavior of metals (particularly the ones selected for this study) in the environment and in biological systems, with reference to their physico-chemical properties. This would smooth the justification of using different metals in this study.

• Further, please incorporate available information on the responses of Azolla plants under metals stress conditions.

• Likewise, the responses of endophytic microbes to metal stress could be incorporated in a way supporting the hypothesis of the study.

We added required information in the Introduction in section starting form line 60.

“Lead, cadmium, chromium and nickel belong to the most widespread heavy metals in environment and are considered as non-essential (highly toxic) trace elements. Because of their long persistence in environment they tend to accumulate in soils, migrate to waters both entering into food chain resulting in concentrations exceeding safety limits which eventually leads to serious health consequences and deterioration soil productivity [13, Dixit et el., 2015]. Noble metals such as gold and silver are also common in environment being important part of jewelry and electronics industry [Burat et al., 2020] and also belong non-essential trace elements having negative effects on living organisms [13, Dixit et al., 2015]. The structure of soil microbial populations is also related to the levels of heavy metals. As a result of metal presence microbial metabolism is affected resulting in lower soil health and fertility [14]. The response of microorganisms depends not only on concentration and availability of metals but also on type of metal, medium and the species present [Dixit et al., 2015].

Due to above mentioned problems there is a need for metal removal from contaminated environments. There are several techniques for coping with heavy metal pollution including chemical precipitation, oxidation or reduction, filtration, ion-exchange, reverse osmosis, membrane technology, evaporation and electrochemical treatment. But most of these techniques become ineffective or expensive when metal concentrations are below 100 mg L-1. In addition, good solubility of metals salts in water make them impossible to be separated with physical methods [Dixit et al., 2015]. Biological methods allowing the use of plant and microorganisms for metal biosorption and/or accumulation are an attractive, efficient and environmentally friendly alternative to physical/chemical methods [13, Dixit et al., 2015]. Among these, the application of plants and microorganisms is both efficient and environmentally friendly [13]. Microorganisms possess many adaptation allowing them to detoxify metals via biosorption, bioaccumulation, biotransformation and biomineralization what is used for different bioremediation methods. Similarly to it plants can be also used for metal remediation which is also very effective solar energy-driven method for metal removal (phytoremediation). The plants having ability to accumulate high levels of metals and growing very fast are called hyperaccumulators and are the best for this purpose. However, there is limited number of such plants and the solution for this may be an aid of microorganisms. As they possess many mechanisms to cope with heavy metals and, as mentioned above, are able to support plants growth (PGPB) their application may be beneficial for successful phytoremediation of metals [Dixit et al., 2015]. For that reason studying microbiomes with regard to metal tolerance is of interest for better designing metal-treatment processes. The process of supporting plants in phytoremediation is called assisted phytoremediation [12,15]. Microorganisms involved in this process are well-adapted to metal-polluted environments, where higher organisms are unable to occur. They have developed capabilities to protect themselves from heavy metal toxicity by various mechanisms such as adsorption, uptake, methylation, oxidation, and reduction [16].

Among hyperaccumulators effective in heavy metal removal by phytoremediation [17] is the aquatic fern Azolla filiculoides L. (Salviniaceae), which possess a recently recognized endophytic microbiome [18,19] composed of interesting microorganisms with PGPB potential [19]. The plant was tested in numerous studies for the potential of the removal many metals among which Pb, Cd, Cr, Ni, Ag and Au are often studied [17,20,73,78,79].”

Dixit, R., Wasiullah, Malaviya, D., Pandiyan, K., Singh, U.B., Sahu A., Shukla, R., Singh, B.P., Rai, J.P., Sharma, P.K., Lade, H., Paul, D. 2015. Bioremediation of Heavy Metals from Soil and Aquatic Environment: An Overview of Principles and Criteria of Fundamental Processes. Sustainability, 7: 2189-2212; doi:10.3390/su7022189

• Is Azolla filiculoides a metal hyperaccumulator (L21)? Related information and reference can be incorporated in the Introduction section to support this statement.

This issue is addressed in the literature reference no. 17 cited in line 72.

• According to sentence in L74-76, authors seem to indicate a double objective for current study i.e (i) to assess the tolerance of Azolla microbiome, (ii) to assess their role in assisted phytoremediation. Has the second objective been investigated?

We agree that the second objective was not here investigated, we can only assume that the presence of metal-tolerant species have some impact on Azolla performance. For that reason we changed next sentence as follows: “To answer the first question…”. We addressed the second in conclusions as a future studies on recognized microorganisms (last sentence).

4. In the Materials and Methods section, authors should indicate the basis dictating the selection of different metals retained for this study.

We added in line 97: “The selection of metals was based on our previous studies with Azolla sp. [20]. These metals constitute environmental risk and their mitigation is often studied.”

• L107: is it 3 or 2 g portion of the sample?

Here we mean that from each Azolla sp. experiment we collected 3 samples each of 2 g – “Three portions of each sample (2 g) …”

• L102: Which part of the plant was sampled? Were plants materials collected to make bulked samples? Please precise. It is commonly believed that the abundance of endophytes varies with regard to plant part. If that is true for Azolla, can authors ascertain whether the procedure used for samples collection didn’t influence results on endophytes abundance?

We conducted severe sterilization procedure to avoid isolation any epiphytic microorganisms. To make a bulk sample we collected material randomly, averaged it and used for DNA isolation. As A. filiculoides is a tiny plant we decided to make isolation from whole plant without separation into shoots and roots. However, it is a good point for future study to obtain endophytes from different parts of A. filiculoides. We believe that obtained results represent overall A. filiculoides microbiome what is sufficient to assess the role of metals on its composition.

5. Authors reported higher richness of endophytes under Pb, Cd and Cr-treatments than Ni, Au and Ag treatments. It is of great interest to discuss the possible (s) raison (s) behind such differential effects of metals.

We agree that these results are interesting and need to be further discusses. We added: “It could be noticed that most of recognized metal-tolerant bacteria showed potential for the accumulation of Pb, Cd and Cr. It could mean that they possess mechanisms to tolerate these metals the most and this could be the reason that the highest abundances were noted in Pb-, Cd- and Cr-treatments”.

Reviewer #2

General comments:

The present manuscript entitled " The endophytic microbiome of Azolla filiculoides L. exposed to selected heavy metals” contains a good idea and impressive amount of work to explain the tolerance ability of target fern (in combination with microbiome) exposed to selected heavy metals. This research is problem-based and might help scientists working on phytoremediation or bioremediation. Nevertheless, authors need some improvement in the manuscript for their possible acceptance in the Journal. Following are some suggestions to be incorporated before making any decision. 

Specific Comments:

Abstract section:

1. Make clear hypothesis at the start of this section.

We changed the beginning of the abstract to emphasize our hypothesis by changing second sentence into: “The metal hyperaccumulator Azolla filiculoides is accompanied by a microbiome potentially supporting plant during exposition to heavy metals. We hypothesized that the microbiome exposition to selected heavy metals will reveal metal tolerant strains”.

2. Overall it is fine

Introduction section:

3. In line 42-43: Rephrase the sentence. Also, delete the extra addition of “and” and which” from these lines.

We changed the sentence as follows: “Plants are one of the most important hosts for complex communities of microorganisms colonizing plant tissues (endosphere – endophytes), outer plant surfaces (phyllosphere – epiphytes) and root surfaces (rhizosphere) forming the plant microbiota [1].”

4. In line 47-48: Delete extra commas (,)

We deleted not necessary commas from lines 47-48.

5. In line 53-56: Rewrite the sentence to make it clearer. Avoid repetition of words

We changed the sentence into: “Plant-associated microorganisms produce substances such as phytohormones and antibiotics which support plant growth and provide protection against pathogens (Plant Growth Promoting Bacteria, PGPB) [4,8,9].”

6. In line 72: “One plant” should be “Among aquatic ferns” or “Among hyperaccumulators” and “heavy metal phytoremediation” should be “heavy metal removal by phytoremediation”, change accordingly.

We change the sentence in line 73 according to suggestion of the reviewer: “Among hyperaccumulators effective in heavy metal removal by phytoremediation [17] is the aquatic fern Azolla filiculoides L. (Salviniaceae), which possess a recently recognized endophytic microbiome [18,19] composed of interesting microorganisms with PGPB potential [19].”

7. In line 75: Mention the particular heavy metals

We named the particular heavy metals: “…their tolerance to heavy metals such as Pb, Cd, Cr, Ni, Ag and Au and possible…” and removed them from next sentence changing it as follows: “To answer this question, we exposed A. filiculoides to these metals; the removal….”

8. In line 77-78: Clearly write the sentence i.e. “their………microorganisms”

We re-written the sentence as follows: “…their dosage were based on literature information about Azolla sp. tolerance to these metals”

9. It is suggested to add few sentences describing the phytotoxic impacts of selected heavy metals and their removal by phytoremediation strategies.

We wrote more information of the impacts of studied metals and their removal by adding text from line 60.

“Lead, cadmium, chromium and nickel belong to the most widespread heavy metals in environment and are considered as non-essential (highly toxic) trace elements. Because of their long persistence in environment they tend to accumulate in soils, migrate to waters both entering into food chain resulting in concentrations exceeding safety limits which eventually leads to serious health consequences and deterioration soil productivity [13, Dixit et el., 2015]. Noble metals such as gold and silver are also common in environment being important part of jewelry and electronics industry [Burat et al., 2020] and also belong non-essential trace elements having negative effects on living organisms [13, Dixit et al., 2015]. The structure of soil microbial populations is also related to the levels of heavy metals. As a result of metal presence microbial metabolism is affected resulting in lower soil health and fertility [14]. The response of microorganisms depends not only on concentration and availability of metals but also on type of metal, medium and the species present [Dixit et al., 2015].

Due to above mentioned problems there is a need for metal removal from contaminated environments. There are several techniques for coping with heavy metal pollution including chemical precipitation, oxidation or reduction, filtration, ion-exchange, reverse osmosis, membrane technology, evaporation and electrochemical treatment. But most of these techniques become ineffective or expensive when metal concentrations are below 100 mg L-1. In addition, good solubility of metals salts in water make them impossible to be separated with physical methods [Dixit et al., 2015]. Biological methods allowing the use of plant and microorganisms for metal biosorption and/or accumulation are an attractive, efficient and environmentally friendly alternative to physical/chemical methods [13, Dixit et al., 2015]. Among these, the application of plants and microorganisms is both efficient and environmentally friendly [13]. Microorganisms possess many adaptation allowing them to detoxify metals via biosorption, bioaccumulation, biotransformation and biomineralization what is used for different bioremediation methods. Similarly to it plants can be also used for metal remediation which is also very effective solar energy-driven method for metal removal (phytoremediation). The plants having ability to accumulate high levels of metals and growing very fast are called hyperaccumulators and are the best for this purpose. However, there is limited number of such plants and the solution for this may be an aid of microorganisms. As they possess many mechanisms to cope with heavy metals and, as mentioned above, are able to support plants growth (PGPB) their application may be beneficial for successful phytoremediation of metals [Dixit et al., 2015]. For that reason studying microbiomes with regard to metal tolerance is of interest for better designing metal-treatment processes. The process of supporting plants in phytoremediation is called assisted phytoremediation [12,15]. Microorganisms involved in this process are well-adapted to metal-polluted environments, where higher organisms are unable to occur. They have developed capabilities to protect themselves from heavy metal toxicity by various mechanisms such as adsorption, uptake, methylation, oxidation, and reduction [16].

Among hyperaccumulators effective in heavy metal removal by phytoremediation [17] is the aquatic fern Azolla filiculoides L. (Salviniaceae), which possess a recently recognized endophytic microbiome [18,19] composed of interesting microorganisms with PGPB potential [19]. The plant was tested in numerous studies for the potential of the removal many metals among which Pb, Cd, Cr, Ni, Ag and Au are often studied [17,20,73,78,79].”

10. Further improve the study objectives at the end of this section.

We changed last parts of objective section: “It was hypothesized that the exposition of A. filiculoides to Pb, Cd, Cr(VI), Ni, Au(III) and Ag will affect the microbiome structure depending on metal. The resulting microbiome composition will display various microbial groups of different tolerance to selected metals and the most abundant groups would be identified as metal tolerant species”.

Materials and methods section:

11. In line 98: The selection of particular heavy metals and their doses are not clearly described. Please explain them within the text or provide reference (not unpublished data).

We added more information on the selection of metal doses: “Metal doses were selected as follows: Pb 500 mg L-1, Cd 5 mg L-1, Cr(VI) 100 mg L-1, Ni 100 mg L-1, Au(III) 5 mg L-1 and Ag 5 mg L-1 (Antunes et al., 2001; Arora et al., 2004; Elmachliy et al., 2011; Zazouli and Balark 2015)”. We added following references:

1. Arora, A., A. Sood, and P.K. Singh. 2004. Hyperaccumulation of cadmium and nickel by Azolla species. Indian Journal of Plant Physiology 3: 302–304.

2. Antunes, A.P.M., Watkins, G.M., Duncan, J.R. 2001. Batch studies on the removal of gold (III) from aqueous solution by Azolla filiculoides. Biotechnology Letters 23: 249–251. https://doi.org/10.1023/A:1005633608727

3. Elmachliy, S., Chefetz, B., Tel-Or, E., Vidal, L., Canals, A., Gedanken, A. 2011. Removal of Silver and Lead Ions from Water Wastes Using Azolla filiculoides, an Aquatic Plant, Which Adsorbs and Reduces the Ions into the Corresponding Metallic Nanoparticles Under Microwave Radiation in 5 min. Water Air Soil Pollut 218: 365–370. https://doi.org/10.1007/s11270-010-0650-3

4. Zazouli, M.A., Balark, D. 2015. Removal of hexavalent chromium from aqueous environments using adsorbents (Lemna and Azolla): An Equilibrium and Kinetics Study. Hormozgan Medical Journal 19(2): 127-139. 

12. In line 102: The collection of plant material was carried out after one week, specify the reason?

We changed this sentence as follows: “All samples were exposed to metals for one week which we thought to be sufficient to affect the microbiome of A. filiculoides. After this time plant material was collected and used for isolation of endophytic microorganisms.”

13. In line 149-151: Re-arrange the sentence

We changed the sentence as follows: “The identified sequences are available under accession number PRJNA589741 in the GenBank database (NCBI, https://www.ncbi.nlm.nih.gov/bioproject/589741).”

Results section:

14. In line 185 and so on: Please cite all figure numbers inside the text (not with headings).

Dear Reviewer, we placed the figure captions in the manuscript text according to ‘Submission Guidelines’ which states “Place figure captions in the manuscript text in read order, immediately following the paragraph where the figure is first cited. Do not include captions as part of the figure files or submit them in a separate document.”

15. In line 244: Delete either “percentage” or “%”.

We agree and adopted this part as suggested: “Unidentified microorganisms, belonging to 4 families, constituted 0.3% of this class…”

16. In line 247: Use “%” with last number/value only.

We agree and adopted this part as suggested: “…its abundance of 0.05, 0.089, and 0.012%, respectively…”

17. In line 267: First use “lead (Pb)” and then “Pb” (also for other metals) throughout the section. Be consistent

We changed all names of metals into their symbol in this section as suggested by the Reviewer:

“…confirmed in Pb- and Cr-treated microbiomes reaching a share of 12.55 and 2%, respectively. (…) in +Au treatment. Ag treatment also revealed the presence of this genus at a level of 12.69%. (…)” and in line 343 – “The Ag-treated…”

18. In line 352: Correct as “―gp3”

We corrected this symbol as suggested by the Reviewer.

19. Explanation of results is fine.

Thank you very much for this opinion.

Discussion section:

20. In line 407: “A. filiculoides” should be Italic and write genus “Azolla” due to first appearance.

Thank you very much for noticing this mistake. We corrected it as suggested. (Azolla filiculoides)

21. In line 410: “Only Pb, Cd and Cr resulted in higher abundances of taxa”, Please check “Ag” also showing significant abundance such as in figure 1.

Dear Reviewer, as it can be seen from Fig. 1 Pb, Cd and Cr treatments show much higher abundances as for: Actinobacteria, Bacteroidetes and Firmicutes. In case of Ag-treatment the share of these phyla much lower than Pb, Cd and Cr. These data are supported by richness numbers presented in Table 1. For that reason we believe that our statement is true.

22. In line 465-466: Re-write the sentence “In 1983…………microorganisms”

We changed this sentence as follows: “Babich and Stotzky [43] demonstrated that the growth of Asticcacaulis excentricus was inhibited at 10 ppm of Ni and completely inhibited at 40-50 ppm of Ni.”

23. In line 507:” which” should be replaced with “whose” and “was the highest” should be “was highest”. Please re-check in other sections

We check whole manuscript for these issues and corrected them accordingly.

24. Authors are suggested to further discuss this section with more possible mechanistic approaches. There are some related articles which can help the authors to understand the similar topic; Biologia Plantarum, 2014, 58(1), 131-138; Ecotoxicology and Environmental Safety, 2014, 102, 25-33; and Plant Physiology and Biochemistry, 2019, 145, 142-152.

We extended the similar information written from line 585 by adding information from suggested publications.

Conclusion section:

25. At the end of this section, it is suggested to add one or two more sentences concerning the future prospects of the present study.

We changed the conclusion by removing last sentence and by: “In the future, it will be interesting to further study detected metal-tolerant microorganisms to know their potential in both metal removal and promoting Azolla sp. growth under exposition to heavy metals. In addition, the identification genes and mechanisms involved in metal tolerance by the bacteria will provide important knowledge useful for their future application in metal bioremediation not only in the co-operation with Azolla sp. but also to support metal phytoremediation by other aquatic plants.”

26. Overall, there are some grammatical errors throughout the manuscript. All sections of the present study were well enough explained and discussed (although need correction/improvement). 

We tried to improve the language but the person responsible for grammar is British native so we trust his expertise.

---

## [Decision Letter · Decision Letter 1]

21 Apr 2020

Azolla filiculoides L. as a source of metal-tolerant microorganisms

PONE-D-20-02088R1

Dear Dr. Marek,

We are pleased to inform you that your manuscript has been judged scientifically suitable for publication and will be formally accepted for publication once it complies with all outstanding technical requirements.

With kind regards,

Basharat Ali, Ph.D

Academic Editor

PLOS ONE

Additional Editor Comments (optional):

Reviewers' comments:

Reviewer's Responses to Questions

**Comments to the Author**

1. If the authors have adequately addressed your comments raised in a previous round of review and you feel that this manuscript is now acceptable for publication, you may indicate that here to bypass the “Comments to the Author” section, enter your conflict of interest statement in the “Confidential to Editor” section, and submit your "Accept" recommendation.

Reviewer #1: All comments have been addressed

Reviewer #2: All comments have been addressed

2. Is the manuscript technically sound, and do the data support the conclusions?

Reviewer #1: Yes

Reviewer #2: Yes

3. Has the statistical analysis been performed appropriately and rigorously? 

Reviewer #1: Yes

Reviewer #2: Yes

4. Have the authors made all data underlying the findings in their manuscript fully available?

Reviewer #1: Yes

Reviewer #2: Yes

5. Is the manuscript presented in an intelligible fashion and written in standard English?

Reviewer #1: Yes

Reviewer #2: Yes

6. Review Comments to the Author

Reviewer #1: The manuscript has been substantially improved and meets standard for publication in PONE. I would recommend acceptance in its current form.

Reviewer #2: Authors have improved their manuscript as suggested. Therefore, it can be accepted for possible publication.

7. PLOS authors have the option to publish the peer review history of their article (what does this mean?). If published, this will include your full peer review and any attached files.

Reviewer #1: No

Reviewer #2: No

---

## [Editor Report · Acceptance letter]

23 Apr 2020

PONE-D-20-02088R1 

*Azolla filiculoides* L. as a source of metal-tolerant microorganisms 

Dear Dr. Banach:

I am pleased to inform you that your manuscript has been deemed suitable for publication in PLOS ONE. Congratulations! Your manuscript is now with our production department. 

With kind regards,

on behalf of

Dr. Basharat Ali 

Academic Editor

PLOS ONE